# Theoretical Aspects of Bias and Diversity in Minimum Bayes Risk Decoding

## Abstract

Text generation commonly relies on greedy and beam decoding that limit the search space and degrade output quality. Minimum Bayes Risk (MBR) decoding can mitigate this problem by utilizing automatic evaluation metrics and model-generated pseudo-references. Previous studies have conducted empirical analyses to reveal the improvement by MBR decoding, and reported various observations. However, despite these observations, the theoretical relationship between them remains uncertain. To address this, we present a novel theoretical interpretation of MBR decoding from the perspective of bias-diversity decomposition. We decompose errors in the estimated quality of generated hypotheses in MBR decoding into two key factors: *bias*, which reflects the closeness between utility functions and human evaluations, and *diversity*, which represents the variation in the estimated quality of utility functions. Our theoretical analysis reveals the difficulty in simultaneously improving both bias and diversity, and highlights the effectiveness of increasing diversity to enhance MBR decoding performance. This analysis verifies the alignment between our theoretical insights and the empirical results reported in previous work. Furthermore, to support our theoretical findings, we propose a new metric, pseudo-bias, which approximates the bias term using gold references. We also introduce a new MBR approach, Metric-augmented MBR (MAMBR), which increases diversity by adjusting the behavior of utility functions without altering the pseudo-references. Experimental results across multiple NLP tasks show that the decomposed terms in the bias-diversity decomposition correlate well with performance, and that MAMBR improves text generation quality by modifying utility function behavior. Our code will be available at `https://github.com/[Anonymized]`.

## 1 Introduction

As demonstrated by the success of large language models (LLMs) (Brown et al., 2020; OpenAI et al., 2024), text generation is one of the most fundamental tasks in Natural Language Processing (NLP). Text generation commonly relies on greedy and beam searches, which heavily restrict the search space when decoding texts from a model. This procedure, which only considers the model's predictions within a limited search space, can sometimes degrade the quality of the generated text.

Minimum Bayes Risk (MBR) decoding (Goel & Byrne, 2000) can mitigate this problem by using a utility function, essentially an automatic evaluation metric, along with pseudo-references generated by the model. MBR decoding was initially applied to speech recognition (Goel & Byrne, 2000) and later to statistical machine translation (SMT) (Kumar & Byrne, 2002; 2004; Duan et al., 2011). Following these successes, MBR decoding has been expanded to various text generation tasks, including neural machine translation (NMT) (Stahlberg et al., 2017), text summarization (Bertsch et al., 2023), and image captioning (Borgeaud & Emerson, 2020).

Since MBR decoding has become an important inference technique in text generation, various empirical studies have explored its characteristics. Müller & Sennrich (2021); Freitag et al. (2022a); Fernandes et al. (2022); Amrhein & Sennrich (2022) highlight the importance of using high quality evaluation metrics that is robust and correlate well with human evaluations as utility functions. Jinnai et al. (2024a); Heineman et al. (2024) emphasize the importance of high-quality pseudo-references that closely resemble human-created ones, while also stressing the significance of pseudo-reference

diversity. Although these empirical findings cover various aspects in detail, a unified interpretation remains challenging due to the lack of theoretical frameworks explaining the relationships behind them.

To address this gap, we provide theoretical interpretations of MBR decoding through bias-diversity decomposition (Krogh & Vedelsby, 1994; Wood et al., 2024). Our theoretical interpretation focuses on errors in the estimated quality of hypotheses in MBR decoding. These errors are decomposed into two critical factors: *bias* and *diversity*. The bias term represents the closeness between the estimated quality produced by utility functions and human evaluations. The diversity term reflects the variance in the estimated quality across different utility functions. Based on this interpretation, we theoretically demonstrate the difficulty in improving both the bias and diversity terms simultaneously, and we highlight the effectiveness of increasing diversity in MBR decoding, verifying the correspondence with empirically induced results from previous work.

To empirically verify our theoretical findings, we propose *pseudo-bias*, which approximates the bias term using gold references. Furthermore, to explore the potential for increasing the diversity term by adjusting the behavior of utility functions without varying the pseudo-references, we introduce a new MBR approach: Metric-augmented MBR (MAMBR).

Our empirical analysis on machine translation, text summarization, and image captioning—using pseudo-references generated by five different sampling methods—shows that the decomposed bias and diversity terms correlate with performance, consistent with our theoretical analysis. Moreover, using MAMBR demonstrates performance improvements by simply modifying the behavior of utility functions.

## 2 MINIMUM BAYES RISK (MBR) DECODING

Minimum Bayes Risk (MBR) decoding (Goel & Byrne, 2000) estimates the quality of each hypothesis $h$ in a set $\mathcal{H}$ by comparing it with each pseudo-reference (evidence sample) $y$ in a set of all sequences $\Omega$ and its model predicted probability $P(y|x)$ for a given input sequence $x$. This process uses an evaluation metric, treated as a utility function $f_\theta(h, y)$, which calculates the similarity between $h$ and $y$ to choose the best hypothesis $\hat{h}_{best}$ in $\mathcal{H}$ as follows:

$$\hat{h}_{best} = \arg\max_{h \in \mathcal{H}} \sum_{y \in \Omega} f_\theta(h, y) P(y|x), \tag{1}$$

where $\theta$ represents the parameters of the evaluation metric used as the utility function $f_\theta(h, y)$. Since calculating $\Omega$ is intractable, Eikema & Aziz (2020; 2022) replace $\Omega$ with $|\mathcal{Y}|$, a set of sampled $y$ as follows:

$$\hat{h}_{mbr} = \arg\max_{h \in \mathcal{H}} \frac{1}{|\mathcal{Y}|} \sum_{y \in \mathcal{Y}} f_\theta(h, y), \quad y \sim P(y|x). \tag{2}$$

Here, instead of using the utility function $f_\theta(h, y)$, we can assume human-estimated quality (Naskar et al., 2023; Suzgun et al., 2023; Jinnai et al., 2024a; Ohashi et al., 2024) denoted as $\hat{f}_{\hat{\theta}}(h)$. Under this assumption, the ideal decoding, which fully relies on human-estimated results, is represented as follows:

$$\hat{h}_{human} = \arg\max_{h \in \mathcal{H}} \hat{f}_{\hat{\theta}}(h). \tag{3}$$

In this paper, we focus on analyzing the differences between the internally estimated qualities for each hypothesis by MBR decoding and those estimated by humans to better understand the characteristics of MBR decoding (§3).

## 3 THEORETICAL ANALYSIS BASED ON BIAS-DIVERSITY DECOMPOSITION

### 3.1 EVALUATION DISCREPANCY

To measure the discrepancy between the human estimated quality, $\hat{f}_{\hat{\theta}}(h)$ and the MBR decoding estimated quality, $\frac{1}{|\mathcal{Y}|} \sum_{y \in \mathcal{Y}} f_\theta(h, y)$, we define a $|\mathcal{H}|$-dimensional vector $\mathbf{u}^j$ that represents estimated

quality for each hypothesis based on the $j$-th pseudo-reference and also define $\bar{\mathbf{u}}$, the average vector of all $\mathbf{u}^j$ as follows:

$$\mathbf{u}^j = \begin{bmatrix} u_1^j \\ \cdots \\ u_{|\mathcal{H}|}^j \end{bmatrix}, \quad u_i^j = f_\theta(h_i, y_j), \quad \bar{\mathbf{u}} = \frac{1}{|\mathcal{Y}|} \sum_{j=1}^{|\mathcal{Y}|} \mathbf{u}^j. \tag{4}$$

Similarly, we can define a $|\mathcal{H}|$-dimensional vector, $\hat{\mathbf{u}}$ that represents the human estimated quality for each hypothesis as follows:

$$\hat{\mathbf{u}} = \begin{bmatrix} \hat{u}_1 \\ \cdots \\ \hat{u}_{|\mathcal{H}|} \end{bmatrix}, \quad \hat{u}_i = \hat{f}_{\hat{\theta}}(h_i). \tag{5}$$

Here, by using Equations 4 and 5, we can reformulate MBR decoding in Equation 2 and the ideal decoding in Equation 3 as follows:

$$(2) \equiv \hat{h}_{mbr} = \arg\max_{h_i} \bar{u}_i, \quad (3) \equiv \hat{h}_{human} = \arg\max_{h_i} \hat{u}_i. \tag{6}$$

Therefore, based on Equation 6, we can investigate the discrepancy between the estimated quality by MBR decoding and human through the comparison of $\bar{\mathbf{u}}$ and $\hat{\mathbf{u}}$. In our work, to estimate the discrepancy, we consider the prediction error of $\bar{\mathbf{u}}$ to $\hat{\mathbf{u}}$ by using Mean Squared Error (MSE) as follows:

$$MSE(\hat{\mathbf{u}}, \bar{\mathbf{u}}) = \frac{1}{|\mathcal{H}|} \sum_{i=1}^{|\mathcal{H}|} (\hat{u}_i - \bar{u}_i)^2. \tag{7}$$

### 3.2 Bias-diversity Decomposition

Our goal is to reveal the characteristics of MBR decoding through theoretical analysis. To achieve this, we focus on the bias and diversity underlying Equation 7. Based on this approach, we can induce the following decomposition:

**Theorem 1.** *The quality estimation error for each hypothesis in MBR decoding, $(\hat{u}_i - \bar{u}_i)^2$, can be decomposed to bias and diversity (ambiguity) terms (Krogh & Vedelsby, 1994) as follows:*

$$(\hat{u}_i - \bar{u}_i)^2 = \underbrace{\frac{1}{|\mathcal{Y}|} \sum_{j=1}^{|\mathcal{Y}|} (\hat{u}_i - f_\theta(h_i, y_j))^2}_{Bias} - \underbrace{\frac{1}{|\mathcal{Y}|} \sum_{j=1}^{|\mathcal{Y}|} (\bar{u}_i - f_\theta(h_i, y_j))^2}_{Diversity}. \tag{8}$$

*Proof.* See Appendix A. □

In Equation 8, two terms represent bias and diversity. Unlike the well-known bias-variance decomposition (Geman et al., 1992) that targets a single estimator which is $\mathbf{u}$ in our case[1], the second term is negative, which is why it is referred to as diversity rather than variance (Wood et al., 2024). The bias term indicates how closely the utility function's estimated quality for a hypothesis matches human estimation. The diversity term reflects how different the utility function's estimated qualities are for each other. This decomposition emphasizes the importance of increasing the diversity term while reducing the bias term to improve the quality estimation error, $(\hat{u}_i - \bar{u}_i)^2$, for each hypothesis.

While MBR decoding considers all hypotheses to rank and select the best one, Theorem 1 addresses only the quality estimation for each hypothesis. To bridge this gap, we decompose $MSE(\hat{\mathbf{u}}, \bar{\mathbf{u}})$ that accounts for all hypotheses. The following theorem addresses this broader perspective:

**Theorem 2.** *The quality estimation error for all hypotheses in MBR decoding, $MSE(\hat{\mathbf{u}}, \bar{\mathbf{u}})$, can be decomposed into bias and diversity terms for all hypotheses as follows:*

$$MSE(\hat{\mathbf{u}}, \bar{\mathbf{u}}) = \underbrace{\frac{1}{|\mathcal{H}|} \sum_{i=1}^{|\mathcal{H}|} \frac{1}{|\mathcal{Y}|} \sum_{j=1}^{|\mathcal{Y}|} (\hat{u}_i - f_\theta(h_i, y_j))^2}_{Bias \ for \ all \ hypotheses} - \underbrace{\frac{1}{|\mathcal{H}|} \sum_{i=1}^{|\mathcal{H}|} \frac{1}{|\mathcal{Y}|} \sum_{j=1}^{|\mathcal{Y}|} (\bar{u}_i - f_\theta(h_i, y_j))^2}_{Diversity \ for \ all \ hypotheses}. \tag{9}$$

---

[1]This becomes $MSE(\hat{\mathbf{u}}, \mathbf{u}) = \frac{1}{|\mathcal{H}|} \sum_{i=1}^{|\mathcal{H}|} \left( (\hat{u}_i - (\frac{1}{|\mathcal{H}|} \sum_{j=1}^{|\mathcal{H}|} u_j))^2 + ((\frac{1}{|\mathcal{H}|} \sum_{j=1}^{|\mathcal{H}|} u_j) - u_i)^2 \right)$.

*Proof.* See Appendix B. □

As in Theorem 1, Theorem 2 highlights the importance of increasing the diversity term while decreasing the bias term to improve the quality estimation in MBR decoding.

### 3.3 INTERPRETATION

The decompositions presented in Theorems 1 and 2 allow us to provide theoretical interpretations for the empirically analyzed characteristics of MBR decoding and its extensions discussed in prior studies.

#### 3.3.1 CORRELATION TO HUMAN EVALUATION RESULTS

The bias term of the decomposition, $(\hat{u}_i - f_\theta(h_i, y_j))^2$, highlights the importance of considering the closeness between the human-estimated quality, $\hat{u}_i$, and the quality estimated by the utility function, $f_\theta(h_i, y_j)$, for improving the performance of MBR decoding. Specifically, since the utility function, $f_\theta(h_i, y_j)$, is influenced by the pseudo-reference $y_j$, the bias term underscores the significance of considering the utility function's correlation to human evaluation and the closeness between pseudo-references and human-created references. Therefore, it emphasizes the importance of examining both utility functions and sampling strategies for generating pseudo-references.

**Quality of Evaluation Metrics.** Our theoretical insight is supported by empirical findings (Müller & Sennrich, 2021; Freitag et al., 2022a; Fernandes et al., 2022; Amrhein & Sennrich, 2022), in which the quality of evaluation metrics used as utility functions is crucial for performance improvement.

**Quality of Pseudo-References.** Ohashi et al. (2024); Jinnai et al. (2024a) empirically show the importance of selecting appropriate pseudo-references. Thus, our findings theoretically support these empirically driven insights of these previous studies.

**Challenges in the Real World.** Our theoretical findings emphasize the necessity of directly reducing the bias term. However, this requires human evaluation of the combination of pseudo-references and evaluation metrics, used as utility functions, for each hypothesis. This task is clearly challenging due to the high cost of human evaluation. As a solution, we propose a method to approximate this in §4.1 and evaluated its correlation with task-specific performance in §5.3.

#### 3.3.2 DIVERSITY OF AUTOMATIC EVALUATION RESULTS

Based on the diversity term of the decomposition, increasing diversity can contribute to performance improvements by reducing each prediction error $(\hat{u}_i - \bar{u}_i)^2$ of $MSE(\hat{\mathbf{u}}, \bar{\mathbf{u}})$ in MBR decoding. A key insight here is that the diversity expressed by $(\bar{u}_i - f_\theta(h_i, y_j))^2$ stems from the different estimated qualities produced by each utility function $f_\theta(h_i, y_j)$. Thus, this diversity can be influenced by the pseudo-reference $y_j$ and/or the model parameters $\theta$ of the evaluation metric.

**Diversity of Pseudo-references.** This finding supports the previous studies (Freitag et al., 2023a; Jinnai et al., 2024a; Heineman et al., 2024) that conclude the diversity of sampling methods is essential for performance improvement of MBR decoding considering that the diversity of the pseudo-references can indirectly contribute to increasing the diversity of $f_\theta(h_i, y_j)$ by each $y_j$. Basically, as Jinnai et al. (2024a) introduce, reranking algorithms are more effective when the candidates are diverse (Gimpel et al., 2013; Li & Jurafsky, 2016; Li et al., 2016) owing to their diverse information to make a consensus.

**Diversity of Evaluation Metrics.** We anticipate performance improvements by combining multiple different evaluation metrics as utility functions to increase diversity. While this approach has been shown to improve the quality estimation of generated texts (Glushkova et al., 2023), to the best of our knowledge, it has not yet been applied to MBR decoding.

**Unexplored Aspect.** Furthermore, the effect of increasing the diversity of estimated qualities from utility functions by varying the evaluation metric's model parameters $\theta$ remains uncertain. To investigate this, we propose a method to adjust the diversity of estimated qualities by modifying $\theta$ in §4.2 and compare its behavior with that of varying pseudo-references in §5.4.

### 3.3.3 MBR DECODING AS ENSEMBLE LEARNING

Our decomposition of MBR decoding aligns with ensemble learning, which is induced by Krogh & Vedelsby (1994). Thus, we can understand that the quality estimation by MBR decoding is a kind of ensemble learning.

**Quality Estimation.** Our decomposition starts from the definition $MSE(\hat{\mathbf{u}}, \bar{\mathbf{u}})$ in Eq. 7, the error between the estimated qualities from human evaluation and MBR decoding. We can actually observe the reduction of errors as the improvement in quality score estimation of (Naskar et al., 2023; Cheng & Vlachos, 2024) by ensembling utility functions that are similar to MBR decoding.

**Weighted-voting.** Furthermore, this viewpoint supports the validity of the previous work (Suzgun et al., 2023; Bertsch et al., 2023) that shows the interpretation of MBR decoding as soft-weighted voting, a variant of ensemble learning. Different from our setting, soft-weighted voting restricts the value range of voters (utility functions) from 0 to 1. Wood et al. (2024) shows that soft-weighted voting can be converted to the decomposition of Krogh & Vedelsby (1994), equivalent to our decomposition in Eq. 8. Therefore, weighted voting-based MBR decoding is similarly explained in our decomposition.

**Number of Pseudo-references.** Generally, increasing the number of pseudo-references improves performance but demands additional computational cost. DeNero et al. (2009); Eikema & Aziz (2022); Cheng & Vlachos (2023); Deguchi et al. (2024b); Vamvas & Sennrich (2024); Trabelsi et al. (2024) prune samples to speedup inference and maintain the original quality similar to the case of pruning estimators in ensemble learning (Liu et al., 2004; Bonab & Can, 2016; 2019).

Considering an ensemble learning method, such as the Bayes optimal classifier (Mitchell, 1997), and assuming that Eq. 2 approximates the expectation by sampling $y_j$, we can explain the performance improvement of increased pseudo-references by the law of large numbers and the success of the pruning and weighted utility functions (Jinnai et al., 2024b) through importance sampling (Kloek & Van Dijk, 1978). (See Appendix C for more details.)

### 3.3.4 BIAS AND DIVERSITY TRADE-OFF

At first glance, based on the interpretation in §3.3.1 and §3.3.2, decreasing bias while increasing diversity seems to be the best strategy to improve quality estimation performance in MBR decoding, which was investigated by Jinnai et al. (2024a). To understand the validity of this strategy, we need to focus on the bias-diversity trade-off (Krogh & Vedelsby, 1994).

**Limitation of MBR Decoding.** The bias-diversity trade-off highlights the difficulty of decreasing bias while increasing diversity. In Eqs. 8 and 9, when the bias term approaches zero, i.e., each $f_\theta(h_i, y_j)$ approaching to $\hat{u}_i$, the diversity term also approaches zero. This theoretical fact indicates that even if we can prepare high-quality evaluation metrics and high-quality pseudo-references that correlate well with human behavior, there may be no performance improvement due to diminished diversity.

**Diversity Assists Inferior Methods.** Conversely, when the evaluation metrics and pseudo-references are inferior, we can expect performance improvements through increased diversity at the cost of increased bias. This phenomenon can explain the sometimes competitive performance of BLEU (Papineni et al., 2002) against COMET (Rei et al., 2020) in Freitag et al. (2022b), and that of ancestral sampling (Robert, 1999) against other sampling methods in Freitag et al. (2023a); Ohashi et al. (2024) using MBR decoding. However, unlike the case where decreased bias leads to decreased diversity, increased bias does not guarantee increased diversity. Therefore, we must carefully assess their diversity when using low-quality evaluation metrics and pseudo-references in MBR decoding.

## 4 REMAINING PROBLEMS & SOLUTIONS

Our theoretical analysis covers various aspects of MBR decoding. However, for a comprehensive analysis, we should investigate empirical results not addressed in previous work and bridge the gap between theory and real-world applications. To this end, we provide the following solutions.

### 4.1 PSEUDO-BIAS

As discussed in §3.3.1, the bias term suggests the importance of considering the correlation between the results of human evaluation and the evaluation metric's decisions based on pseudo-references to improve the performance of MBR decoding. However, calculating the bias term requires human evaluation, and conducting human evaluations for each setting is unrealistic and difficult. To address this issue, we introduce *pseudo-bias*, an approximation of the bias term in our decomposition. By using $|\hat{\mathcal{Y}}|$, the number of gold references $\hat{y}$, pseudo-bias is defined as follows:

$$\underbrace{\frac{1}{|\mathcal{Y}|} \sum_{j=1}^{|\mathcal{Y}|} (\hat{u}_i - f_\theta(h_i, y_j))^2}_{\text{Bias}} \approx \underbrace{\frac{1}{|\mathcal{Y}|} \sum_{j=1}^{|\mathcal{Y}|} (\widetilde{u}_i - f_\theta(h_i, y_j))^2}_{\text{Pseudo-bias}}, \qquad \widetilde{u}_i = \frac{1}{|\hat{\mathcal{Y}}|} \sum_{j=1}^{|\hat{\mathcal{Y}}|} f_\theta(h_i, \hat{y}_j). \quad (10)$$

This formulation is based on the premise that automatic evaluation metrics correlate to human evaluation when receiving human-created references.[2] Since we can calculate the diversity term without any approximation, we compare pseudo-bias with diversity in terms of how they correlate with performance.

### 4.2 METRIC-AUGMENTED MBR

The discussion in §3.3.2 shows the possibility of increasing the diversity of the utility function, $f_\theta(h_i, y_j)$, by changing the evaluation metric's model parameters, $\theta$, as well as by introducing diversity through pseudo-references. To this end, we propose a new method called Metric-augmented Minimum Bayes Risk (MAMBR) decoding. In MAMBR, we employ different parameters for the evaluation metric to enhance the diversity of utility functions. Letting $\Theta$ be a set of model parameters, MAMBR is defined as follows:

$$\hat{h}_{\text{mambr}} = \arg\max_{h \in \mathcal{H}} \frac{1}{|\mathcal{Y}| |\Theta|} \sum_{\theta \in \Theta} \sum_{y \in \mathcal{Y}} f_\theta(h, y). \quad (11)$$

When using MAMBR, we train evaluation metrics with different initial random seeds to generate $\Theta$ as a set of diverse model parameters.

## 5 EMPIRICAL ANALYSIS

We conduct empirical analysis corresponding to our theoretical analysis through experiments for a comprehensive understanding of MBR decoding.

### 5.1 OVERALL SETTINGS

We target three different text generation tasks, machine translation, text summarization, and image captioning to investigate the general performance of MBR decoding. In all tasks, we followed the settings of Jinnai et al. (2024b) for generating samples. We used epsilon sampling (Hewitt et al., 2022) to generate hypotheses.[3] For the generation of pseudo-references, we used various sampling approaches: beam decoding, nucleus sampling (Holtzman et al., 2020) with $p = 0.9$, ancestral sampling, top-$k$ sampling (Fan et al., 2018) with $k = 10$, and epsilon sampling with $\epsilon = 0.02$. We set the sampling size for hypotheses to 64. We chose the sampling size for pseudo-references from $\{4, 8, 16, 32, 64\}$. We used the following datasets, models[4], and evaluation metrics for each task:

**Machine Translation** We used the WMT19 English to German (En-De) and WMT19 English to Russian (En-Ru) datasets (Barrault et al., 2019). We used `facebook/wmt19-en-de` for En-De

---

[2]For the pseudo-bias, we used COMET (`Unbabel/wmt22-comet-da`) and BERTScore with `microsoft/deberta-xlarge-mnli` whose pearson correlations are 0.990 on the system-level task for English to German (Freitag et al., 2023b) and 0.7781 (`https://github.com/Tiiiger/bert_score`) on WMT16 to English (Bojar et al., 2016), respectively.

[3]Appendix G.1 includes the results with hypotheses generated by different sampling methods.

[4]We used all models from `https://huggingface.co/models` (Wolf et al., 2020).

| Spearman's Rank Correlation | WMT19 En-De | WMT19 En-Ru | SAMSum | XSum | MSCOCO | NoCaps | Avg. |
|---|---|---|---|---|---|---|---|
| Overall Bias | 0.56 | 0.24 | 0.75 | 0.35 | -0.21 | -0.04 | *0.32* |
| One Best Bias | 0.58 | 0.36 | 0.67 | 0.80 | 0.59 | 0.18 | *0.56* |
| Overall Diversity | 0.46 | 0.64 | 0.30 | 0.56 | 0.66 | 0.04 | *0.46* |
| One Best Diversity | 0.44 | 0.64 | -0.04 | 0.39 | 0.16 | -0.45 | *0.21* |
| Overall MSE | 0.76 | 0.60 | 0.87 | 0.68 | 0.36 | 0.15 | *0.62* |
| One Best MSE | 0.79 | 0.65 | 0.67 | 0.79 | 0.56 | 0.16 | *0.64* |

| Pearson's Correlation | WMT19 En-De | WMT19 En-Ru | SAMSum | XSum | MSCOCO | NoCaps | Avg. |
|---|---|---|---|---|---|---|---|
| Overall Bias | 0.06 | -0.15 | 0.78 | 0.17 | 0.05 | 0.04 | *0.20* |
| One Best Bias | 0.10 | -0.04 | 0.63 | 0.76 | 0.33 | -0.11 | *0.32* |
| Overall Diversity | 0.07 | 0.33 | 0.11 | 0.49 | 0.54 | -0.03 | *0.26* |
| One Best Diversity | 0.06 | 0.30 | 0.03 | 0.39 | 0.14 | -0.46 | *0.08* |
| Overall MSE | 0.15 | -0.01 | 0.83 | 0.37 | 0.23 | 0.01 | *0.31* |
| One Best MSE | 0.20 | 0.12 | 0.59 | 0.80 | 0.31 | -0.20 | *0.35* |

Figure 1: Correlation between measures in our decomposition and performance for each dataset. The underlined scores indicate statistically significant results ($p < 0.05$).[7] Note that the italic scores at *Avg.* are not the target of the significance test.

and `facebook/wmt19-en-ru` for En-Ru, respectively. As the utility function and evaluation metric, we used COMET with the model `Unbabel/wmt22-comet-da`.

**Text Summarization** We used the SAMSum (Gliwa et al., 2019) and XSum (Narayan et al., 2018) datasets, and used `philschmid/bart-large-cnn-samsum` and `facebook/bart-large-xsum` for generation in SAMSum and XSum, respectively. As the utility function and evaluation metric, we used BERTScore (Zhang* et al., 2020) with the model `microsoft/deberta-xlarge-mnli`.

**Image Captioning** We used the MSCOCO dataset (Lin et al., 2014) with the split of Karpathy & Fei-Fei (2015) and the NoCaps dataset (Agrawal et al., 2019). We used `Salesforce/blip2-flan-t5-xl-coco` and `Salesforce/blip2-flan-t5-xl` for generation in MSCOCO and NoCaps, respectively. As the utility function and evaluation metric, we used BERTScore with the model `microsoft/deberta-xlarge-mnli`. Since both datasets have multiple references, we report their average scores.

Our implementation of the generation part is based on the released code of Jinnai et al. (2024b)[5] and the MBR decoding part is based on the toolkit, `mbrs` by Deguchi et al. (2024a)[6]. We generate samples on NVIDIA GeForce RTX 3090 and perform MBR decoding on an NVIDIA RTX A6000.

## 5.2 Correlation of Bias and Diversity to Performance

To verify our theoretical decomposition, we investigate the correlation of bias and diversity to performance on each dataset. For this purpose, we approximately compute the bias term by using our pseudo-bias in §4.1. Furthermore, we investigate the importance of whether to consider the entire candidate (Eq. 9) or one best candidate (Eq. 8).

**Settings** We compared the following measures in our decomposition: Overall Bias, the first term of Eq. 9 for all hypotheses; One Best Bias, the first term of Eq. 8 for the one best result by MBR decoding; Overall Diversity, the second term of Eq. 9 for all hypotheses; One Best Diversity, the second term of Eq. 8 for the one best result by MBR decoding; Overall MSE, indicating errors for all hypotheses based on Eq. 9; and One Best MSE, indicating errors for the one best result by MBR decoding based on Eq. 8. For the comparison, we calculated Spearman's rank correlation and Pearson correlation between these measures and the performance based on the results of five different sampling methods with five different sampling sizes on each dataset (see §5.1). Since lower bias and MSE are better for performance, we took their negative values in the correlation calculation. Moreover, we report averaged correlation across all datasets by Fisher z-transformation (Corey et al., 1998).

**Results** Figure 1 shows the correlation between the measures and performance for each dataset. These results show that MSE for both overall and one best results correlates well with the performance for each dataset in Spearman's rank correlation, indicating the importance of considering quality estimation in MBR decoding, as in Eqs. 8 and 9. On the other hand, the decomposed bias

---

[5] `https://github.com/CyberAgentAILab/model-based-mbr`
[6] `https://github.com/naist-nlp/mbrs`
[7] We used Student's t-test (Student, 1908) for both spearman and pearson correlations.

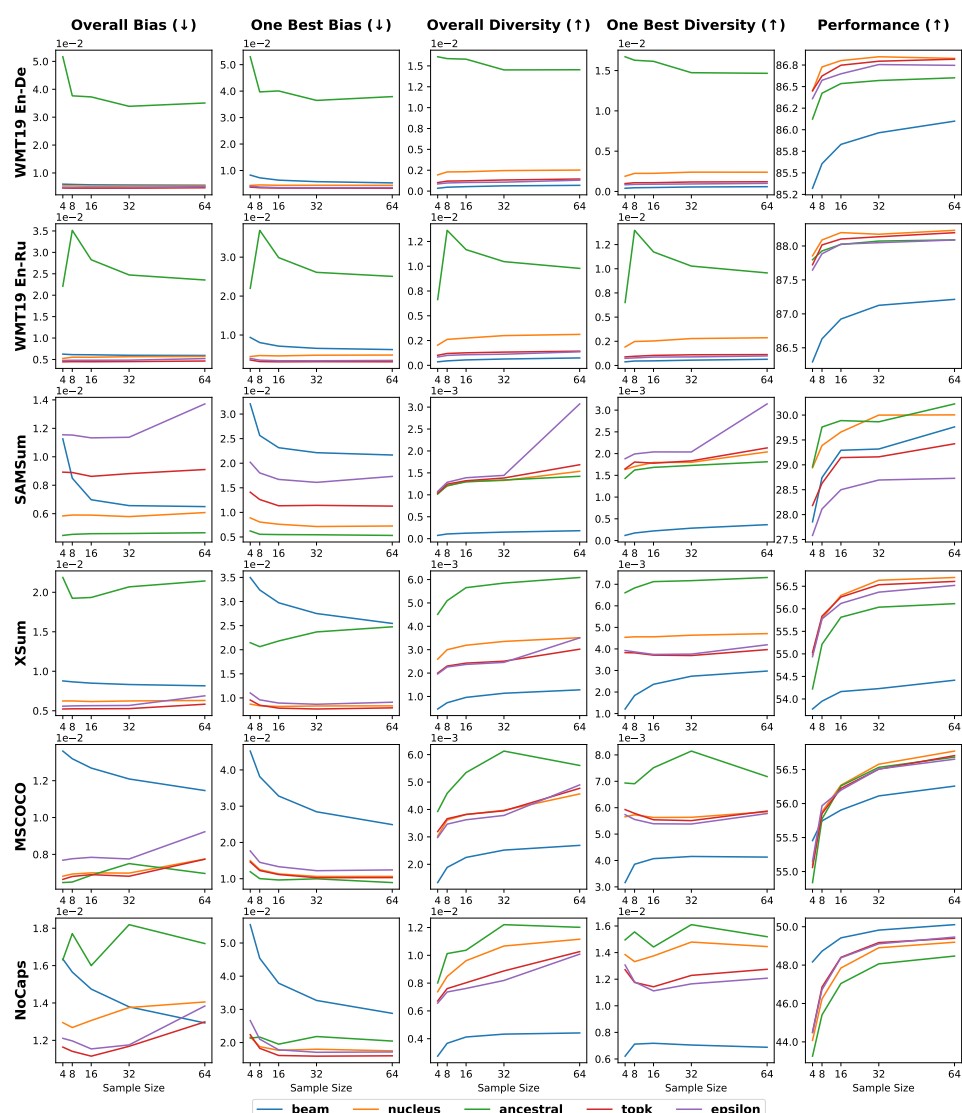

Figure 2: The relationship between bias, diversity, and performance in MBR decoding. The x-axis shows the number of used pseudo-references. (↑) indicates higher scores are better whereas (↓) indicates lower scores are better.

and diversity show different tendencies. ONE BEST BIAS, which considers the one best result, is important for bias, whereas OVERALL DIVERSITY, which considers overall results, is important for diversity. This result is reasonable given an assumption that MBR decoding aims to select texts that are close to human-created ones. Based on this assumption, we can say that diversity supports the selection by considering the importance of all hypotheses not covered by One Best Bias. In contrast to the results in Spearman's rank correlation, the coefficients of Pearson's correlation decrease. Based on these results, we can conclude that the measures, i.e., ONE BEST BIAS, OVERALL DIVERSITY, ONE BEST MSE and OVERALL MSE, correlate well with the rank in performance, but they are difficult to precisely capture subtle differences of values. (See Appendix D for further details.)

## 5.3   BIAS AND DIVERSITY TRADE-OFF

To investigate the bias-diversity trade-off in more detail, we followed the setup described in §5.2. We plotted the results for each dataset using different sampling methods in Figure 2. The results show that while ancestral sampling exhibits the highest bias, except in the case of the SAMSum

Table 1: Results of MAMBR with ancestral sampling. Bold font indicates the best result.

| | | WMT19 En-De | | | | | WMT19 En-Ru | | | | |
|---|---|---|---|---|---|---|---|---|---|---|---|
| Num. of Samples | | 4 | 8 | 16 | 32 | 64 | 4 | 8 | 16 | 32 | 64 |
| | 1 | 85.7 | 85.9 | 85.9 | 85.9 | 85.9 | **87.4** | 87.4 | 87.5 | 87.5 | 87.5 |
| Num. of Models | 2 | 85.7 | **86.0** | **86.0** | 85.9 | 85.9 | **87.4** | 87.4 | 87.5 | 87.5 | **87.6** |
| | 4 | **85.8** | **86.0** | **86.0** | **86.0** | 86.0 | **87.4** | 87.4 | 87.5 | 87.5 | **87.6** |
| | 8 | **85.8** | **86.0** | **86.0** | **86.0** | **86.1** | **87.4** | **87.5** | **87.6** | **87.6** | **87.6** |
| | | SAMSum | | | | | XSum | | | | |
| Num. of Samples | | 4 | 8 | 16 | 32 | 64 | 4 | 8 | 16 | 32 | 64 |
| | 1 | 28.6 | 29.1 | 29.5 | 29.5 | 29.7 | 54.2 | 55.2 | 55.7 | 56.0 | 56.1 |
| Num. of Models | 2 | 28.8 | **29.6** | **29.9** | **29.9** | 30.1 | 54.2 | 55.2 | 55.7 | **56.1** | **56.2** |
| | 4 | **28.7** | 29.5 | **29.9** | 29.8 | **30.2** | 54.2 | 55.2 | **55.8** | **56.1** | **56.2** |
| | 8 | **28.7** | 29.5 | 29.8 | **29.9** | 30.1 | **54.3** | **55.3** | **55.8** | **56.1** | **56.2** |
| | | MSCOCO | | | | | NoCaps | | | | |
| Num. of Samples | | 4 | 8 | 16 | 32 | 64 | 4 | 8 | 16 | 32 | 64 |
| | 1 | **54.9** | 55.8 | 56.3 | 56.5 | 56.8 | 42.9 | 45.3 | 46.8 | 47.8 | 48.6 |
| Num. of Models | 2 | **54.9** | 55.8 | 56.4 | 56.6 | 56.8 | 43.2 | 45.6 | 47.2 | 48.3 | 48.9 |
| | 4 | **54.9** | **56.0** | 56.4 | 56.7 | **56.9** | 43.3 | 45.6 | 47.3 | 48.4 | **49.0** |
| | 8 | **54.9** | **56.0** | **56.5** | **56.8** | **56.9** | 43.5 | 45.7 | 47.4 | 48.5 | 49.0 |

Table 2: Results of MAMBR with epsilon sampling. Notations are the same as Table 1.

| | | WMT19 En-De | | | | | WMT19 En-Ru | | | | |
|---|---|---|---|---|---|---|---|---|---|---|---|
| Num. of Samples | | 4 | 8 | 16 | 32 | 64 | 4 | 8 | 16 | 32 | 64 |
| | 1 | 85.9 | 86.1 | 86.2 | 86.2 | 86.2 | 87.3 | 87.6 | **87.7** | **87.7** | 87.7 |
| Num. of Models | 2 | **86.0** | 86.1 | 86.1 | 86.2 | 86.3 | 87.3 | 87.6 | **87.7** | **87.7** | 87.7 |
| | 4 | **86.0** | 86.1 | 86.2 | 86.2 | 86.3 | **87.4** | 87.6 | **87.7** | **87.7** | 87.7 |
| | 8 | **86.0** | **86.2** | **86.3** | **86.3** | **86.4** | **87.4** | **87.7** | **87.7** | **87.7** | **87.8** |
| | | SAMSum | | | | | XSum | | | | |
| Num. of Samples | | 4 | 8 | 16 | 32 | 64 | 4 | 8 | 16 | 32 | 64 |
| | 1 | 27.5 | 27.9 | 28.3 | 28.4 | 28.5 | 54.9 | 55.7 | 56.1 | 56.3 | 56.4 |
| Num. of Models | 2 | **27.7** | 28.1 | 28.5 | **28.6** | **28.7** | 54.9 | 55.7 | 56.1 | 56.3 | **56.5** |
| | 4 | **27.7** | **28.2** | 28.5 | **28.6** | 28.6 | 54.9 | 55.7 | 56.1 | **56.4** | **56.5** |
| | 8 | 27.6 | **28.2** | **28.6** | **28.6** | **28.7** | 54.9 | **55.8** | **56.2** | **56.4** | **56.5** |
| | | MSCOCO | | | | | NoCaps | | | | |
| Num. of Samples | | 4 | 8 | 16 | 32 | 64 | 4 | 8 | 16 | 32 | 64 |
| | 1 | 55.2 | 55.9 | **56.3** | 56.5 | 56.7 | 44.4 | 46.7 | 48.5 | 49.1 | 49.5 |
| Num. of Models | 2 | 55.2 | 55.9 | **56.3** | 56.5 | 56.7 | 44.4 | 46.8 | 48.6 | 49.2 | 49.6 |
| | 4 | 55.2 | 56.0 | **56.3** | **56.6** | **56.8** | 44.5 | 46.9 | **48.7** | 49.3 | **49.7** |
| | 8 | **55.3** | **56.1** | **56.3** | **56.6** | **56.8** | 44.6 | 47.0 | **48.7** | 49.4 | **49.7** |

dataset, it sometimes outperforms other sampling methods owing to its greater diversity. Focusing on top-k sampling, which has the lowest bias, again excluding the SAMSum dataset, we can observe that the reduction in bias tends to limit the increase in diversity. This finding supports our previously noted bias-diversity trade-off in MBR decoding. However, as evidenced by the performance of beam decoding, which has the lowest diversity, the importance of bias and diversity varies depending on the target dataset. Therefore, while our theoretical analysis effectively explains the performance tendencies in MBR decoding, it remains essential to consider task-specific features carefully to achieve further performance improvements. (See Appendix E for further details.)

## 5.4 EFFECTIVENESS OF METRIC-AUGMENTED MBR

We investigate the possibility of improving performance of MAMBR in Eq. 11 by changing automatic evaluation metric's model parameters.

Table 3: Results of MAMBR with beam decoding. Notations are the same as Table 1.

| | | WMT19 En-De | | | | | WMT19 En-Ru | | | | |
|---|---|---|---|---|---|---|---|---|---|---|---|
| Num. of Samples | | 4 | 8 | 16 | 32 | 64 | 4 | 8 | 16 | 32 | 64 |
| | 1 | 85.2 | 85.4 | 85.6 | 85.7 | 85.8 | 86.5 | 86.8 | **87.0** | 87.1 | 87.1 |
| Num. of Models | 2 | **85.3** | **85.5** | **85.7** | **85.8** | 85.8 | 86.5 | 86.8 | 86.9 | **87.1** | 87.1 |
| | 4 | **85.3** | **85.5** | **85.7** | **85.8** | 85.8 | **86.6** | **86.9** | **87.0** | **87.1** | **87.2** |
| | 8 | **85.3** | **85.5** | **85.7** | **85.8** | **85.9** | 86.5 | 86.8 | **87.0** | **87.1** | **87.2** |
| | | SAMSum | | | | | XSum | | | | |
| Num. of Samples | | 4 | 8 | 16 | 32 | 64 | 4 | 8 | 16 | 32 | 64 |
| | 1 | 27.6 | 28.7 | **29.2** | 29.3 | **29.7** | 53.8 | 54.0 | 54.2 | 54.2 | 54.4 |
| Num. of Models | 2 | **27.8** | **28.9** | 29.2 | **29.4** | **29.7** | 53.8 | 53.9 | 54.1 | **54.2** | **54.4** |
| | 4 | **27.8** | **28.9** | 29.2 | **29.4** | **29.7** | 53.8 | **54.0** | **54.2** | **54.2** | **54.4** |
| | 8 | **27.8** | **28.9** | 29.2 | **29.4** | **29.7** | 53.8 | **54.0** | **54.2** | **54.2** | **54.4** |
| | | MSCOCO | | | | | NoCaps | | | | |
| Num. of Samples | | 4 | 8 | 16 | 32 | 64 | 4 | 8 | 16 | 32 | 64 |
| | 1 | 55.4 | 55.7 | **55.9** | 56.1 | 56.3 | 48.2 | 48.8 | 49.4 | **49.9** | 50.2 |
| Num. of Models | 2 | 55.4 | **55.8** | **55.9** | 56.1 | 56.3 | 48.2 | 48.8 | 49.4 | **49.9** | **50.3** |
| | 4 | **55.5** | 55.7 | **55.9** | 56.1 | 56.3 | 48.2 | 48.8 | **49.5** | **49.9** | 50.2 |
| | 8 | **55.5** | 55.7 | **55.9** | 56.1 | 56.3 | 48.2 | 48.8 | **49.5** | **49.9** | 50.2 |

**Settings**   To prepare the set of model parameters, we trained eight models by varying their initial seeds. We trained `Unbabel/wmt22-comet-da` on the Direct Assessments (DA) task (Graham et al., 2013), using the WMT 2017 to 2020 datasets (Bojar et al., 2017; 2018; Barrault et al., 2019; 2020) for training and the WMT 2021 dataset (Akhbardeh et al., 2021) for validation in COMET. Additionally, we trained `microsoft/deberta-large` on the MNLI dataset from GLUE (Wang et al., 2018) for BERTScore. During inference, to control model diversity, we selected the top-$n$ models based on their proximity to the median validation scores, with $n$ chosen from 1, 2, 4, 8. We aimed to explore the relationship between the diversity caused by pseudo-references and model parameters, focusing on sampling strategies with varying levels of diversity: ancestral sampling, epsilon sampling, and beam decoding.

**Results**   Tables 1, 2, and 3 respectively show the MAMBR results with pseudo-references from ancestral sampling, epsilon sampling, and beam decoding. In ancestral and epsilon sampling, the best and moderately diversified sampling strategies (as shown in Figure 2), we observe performance improvement as the number of models increases. On the other hand, in the lowest diversity method, beam decoding, performance improvement is limited. These results suggest that MAMBR can improve performance by enhancing the diversity of evaluation metrics, although the diversity of the sampling strategy itself remains important. (See Appendix F for further details.)

## 6   CONCLUSION

This work provides a unified theoretical interpretation of Minimum Bayes Risk (MBR) decoding through the lens of bias-diversity decomposition. By decomposing the errors in quality estimation in MBR decoding into bias and diversity, we highlight the trade-off between improving these two factors, with an emphasis on the benefits of increasing diversity. Our theoretical insights align with previous empirical results, and we further investigate aspects not covered by these empirical findings through the introduction of the pseudo-bias metric and the development of the Metric-augmented MBR (MAMBR) approach. Experimental results across multiple tasks demonstrate the validity of our theoretical findings and the effectiveness of our approach in improving text generation quality. These findings bridge the gap between empirical observations and theoretical understanding of MBR decoding, offering new insights for optimizing text generation. (See Appendix H for the limitations of our work.)

## 7 REPRODUCIBILITY STATEMENT

We performed our experiments by running publicly available models, `facebook/wmt19-en-de`, `facebook/wmt19-en-ru`, `philschmid/bart-large-cnn-samsum`, `facebook/bart-large-xsum`, `Salesforce/blip2-flan-t5-xl-coco`, and `Salesforce/blip2-flan-t5-xl` in HuggingFace Transformers (Wolf et al., 2020) on the publicly available datasets, WMT19 English to German (Barrault et al., 2019), WMT19 English to Russian (Barrault et al., 2019), SAMSum (Gliwa et al., 2019), XSum (Narayan et al., 2018), MSCOCO (Lin et al., 2014; Karpathy & Fei-Fei, 2015), and NoCaps (Agrawal et al., 2019), respectively with utilizing the publicly available MBR decoding toolkit, `mbrs` (Deguchi et al., 2024a) as described in §5.1. In addition, we will release our code at `https://github.com/naist-nlp/mbr-bias-diversity`.

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

## A  Proof for Theorem 1

$$(\hat{u}_i - \bar{u}_i)^2 \tag{12}$$

$$=(\hat{u}_i)^2 - 2\hat{u}_i\bar{u}_i + (\bar{u}_i)^2 \tag{13}$$

$$=(\hat{u}_i)^2 - 2\hat{u}_i\bar{u}_i + 2(\bar{u}_i)^2 - (\bar{u}_i)^2 \tag{14}$$

$$=(\hat{u}_i)^2 - 2\hat{u}_i\bar{u}_i + 2\bar{u}_i\bar{u}_i - (\bar{u}_i)^2 \tag{15}$$

$$=(\hat{u}_i)^2 - \frac{1}{|\mathcal{Y}|}\sum_{j=1}^{|\mathcal{Y}|}2\hat{u}_i u_i^j + \frac{1}{|\mathcal{Y}|}\sum_{j=1}^{|\mathcal{Y}|}2\bar{u}_i u_i^j - (\bar{u}_i)^2 \tag{16}$$

$$=\frac{1}{|\mathcal{Y}|}\sum_{j=1}^{|\mathcal{Y}|}(\hat{u}_i)^2 - \frac{1}{|\mathcal{Y}|}\sum_{j=1}^{|\mathcal{Y}|}2\hat{u}_i u_i^j + \frac{1}{|\mathcal{Y}|}\sum_{j=1}^{|\mathcal{Y}|}2\bar{u}_i u_i^j - \frac{1}{|\mathcal{Y}|}\sum_{j=1}^{|\mathcal{Y}|}(\bar{u}_i)^2 \tag{17}$$

$$=\frac{1}{|\mathcal{Y}|}\sum_{j=1}^{|\mathcal{Y}|}((\hat{u}_i)^2 - 2\hat{u}_i u_i^j + 2\bar{u}_i u_i^j - (\bar{u}_i)^2) \tag{18}$$

$$=\frac{1}{|\mathcal{Y}|}\sum_{j=1}^{|\mathcal{Y}|}((\hat{u}_i)^2 - 2\hat{u}_i u_i^j + (u_i^j)^2 - (u_i^j)^2 + 2\bar{u}_i u_i^j - (\bar{u}_i)^2) \tag{19}$$

$$=\frac{1}{|\mathcal{Y}|}\sum_{j=1}^{|\mathcal{Y}|}((\hat{u}_i)^2 - 2\hat{u}_i u_i^j + (u_i^j)^2 - ((u_i^j)^2 - 2\bar{u}_i u_i^j + (\bar{u}_i)^2)) \tag{20}$$

$$=\frac{1}{|\mathcal{Y}|}\sum_{j=1}^{|\mathcal{Y}|}((\hat{u}_i - u_i^j)^2 - (\bar{u}_i - u_i^j)^2) \tag{21}$$

$$=\underbrace{\frac{1}{|\mathcal{Y}|}\sum_{j=1}^{|\mathcal{Y}|}(\hat{u}_i - f_\theta(h_i, y_j))^2}_{\text{Bias}} - \underbrace{\frac{1}{|\mathcal{Y}|}\sum_{j=1}^{|\mathcal{Y}|}(\bar{u}_i - f_\theta(h_i, y_j))^2}_{\text{Diversity}} \tag{22}$$

## B  Proof for Theorem 2

$$MSE(\hat{\mathbf{u}}, \bar{\mathbf{u}}) \tag{23}$$

$$=\frac{1}{|\mathcal{H}|}\sum_{i=1}^{|\mathcal{H}|}(\hat{u}_i - \bar{u}_i)^2 \tag{24}$$

$$=\frac{1}{|\mathcal{H}|}\sum_{i=1}^{|\mathcal{H}|}\Big(\frac{1}{|\mathcal{Y}|}\sum_{j=1}^{|\mathcal{Y}|}(\hat{u}_i - f_\theta(h_i, y_j))^2 - \frac{1}{|\mathcal{Y}|}\sum_{j=1}^{|\mathcal{Y}|}(\bar{u}_i - f_\theta(h_i, y_j))^2\Big) \tag{25}$$

$$=\underbrace{\frac{1}{|\mathcal{H}|}\sum_{i=1}^{|\mathcal{H}|}\frac{1}{|\mathcal{Y}|}\sum_{j=1}^{|\mathcal{Y}|}(\hat{u}_i - f_\theta(h_i, y_j))^2}_{\text{Bias for all hypotheses}} - \underbrace{\frac{1}{|\mathcal{H}|}\sum_{i=1}^{|\mathcal{H}|}\frac{1}{|\mathcal{Y}|}\sum_{j=1}^{|\mathcal{Y}|}(\bar{u}_i - f_\theta(h_i, y_j))^2}_{\text{Diversity for all hypotheses}} \tag{26}$$

## C  INTERPRETATION AS ENSEMBLE LEARNING

When $|\mathcal{Y}|$ is large enough to satisfy the law of large numbers, we can induce the following expectation in MBR decoding by using a model's prediction, $P(y|x)$:

$$\arg\max_{h \in \mathcal{H}} \sum_{y \in \Omega} f_\theta(h, y) P(y|x) \tag{27}$$

$$= \arg\max_{h \in \mathcal{H}} \mathbb{E}_{P(y|x)}[f_\theta(h, y)] \tag{28}$$

$$\approx \arg\max_{h \in \mathcal{H}} \frac{1}{|\mathcal{Y}|} \sum_{y \in \mathcal{Y}} f_\theta(h, y), \quad y_1, \cdots, y_{|\mathcal{Y}|} \sim P(y|x) \tag{29}$$

Since this expectation is based on $P(y|x)$, we can understand the importance of increasing the number of pseudo-references to induce a reliable $P(y|x)$.

**Theorem 3.** *When $f_\theta(h, y)$ is normalized as a probability $P_\theta(h|y)$, Equation 27 is equivalent to Bayes Optimal Classifier (BOC) in Mitchell (1997).*

*Proof.* Self-evident by the following reformulation:

$$\arg\max_{h \in \mathcal{H}} \sum_{y \in \Omega} f_\theta(h, y) P(y|x) = \arg\max_{h \in \mathcal{H}} \sum_{y \in \Omega} P_\theta(h|y) P(y|x) \tag{30}$$

$\square$

**Theorem 4.** *When $f_\theta(h, y)$ is normalized as a probability $P_\theta(h|y)$, Equation 29 is equivalent to the Gibbs algorithm in Mitchell (1997) that approximates BOC by sampling.*

*Proof.* Self-evident by the following reformulation:

$$\arg\max_{h \in \mathcal{H}} \frac{1}{|\mathcal{Y}|} \sum_{y \in \mathcal{Y}} f_\theta(h, y), \quad y_1, \cdots, y_{|\mathcal{Y}|} \sim P(y|x) \tag{31}$$

$$= \arg\max_{h \in \mathcal{H}} \sum_{y \in \mathcal{Y}} P_\theta(h|y), \quad y_1, \cdots, y_{|\mathcal{Y}|} \sim P(y|x) \tag{32}$$

$\square$

Hence, we can understand that MBR decoding represented as Eqs. 27 and 29 approximates the ensemble learning method, BOC. In this interpretation, since $P(y|x)$ is a prior of BOC, we can also understand that MBR approximately uses the model-predicted probability as its prior.

When pruning unnecessary $y$ in the BOC formulation of Equation 30, because the sum of $P_\theta(h|y)$ for all $h$ is always 1, we can determine the importance of $y$ based solely on $P(y|x)$. Since we can arbitrarily choose $P(y|x)$ during sampling, we understand that pruning methods select the importance of each $y$ as a prior in BOC. Note that utility functions are not always normalized; therefore, there is a gap between this interpretation and the actual MBR decoding. Addressing this gap remains an open problem.

In practice, directly drawing samples from $P(y|x)$ is intractable. Therefore, we must use approximate search methods, which are commonly influenced by left-to-right decoding and threshold values. These factors can lead to unreachable states and biases, as seen in greedy or beam decoding and other sampling approaches. Letting $P'(y|x)$ denote the model's prediction with the approximate search, we can similarly induce the following expectation:

$$\arg\max_{h \in \mathcal{H}} \frac{1}{|\mathcal{Y}|} \sum_{y \in \mathcal{Y}} f_\theta(h, y), \quad y_1, \cdots, y_{|\mathcal{Y}|} \sim P'(y|x) \tag{33}$$

$$\approx \arg\max_{h \in \mathcal{H}} \mathbb{E}_{P'(y|x)}[f_\theta(h, y)] \tag{34}$$

Unfortunately, due to $P'(y|x)$, Equation 34 deviates from Equation 28. To precisely predict Eq. 27 using samples from $P'(y|x)$, we can consider the following theorem:

**Theorem 5.** *When $|\mathcal{Y}|$ is enough large to satisfy the law of large numbers, by using importance sampling, we can induce Eq. 28 from $P'(y|x)$.*

*Proof.*

$$\arg\max_{h \in \mathcal{H}} \mathbb{E}_{P(y|x)}[f_\theta(h, y)] \tag{35}$$

$$= \arg\max_{h \in \mathcal{H}} \sum_y P(y|x) f_\theta(h, y) \tag{36}$$

$$= \arg\max_{h \in \mathcal{H}} \sum_y P(y|x) f_\theta(h, y) \frac{P'(y|x)}{P'(y|x)} \tag{37}$$

$$= \arg\max_{h \in \mathcal{H}} \sum_y P'(y|x) f_\theta(h, y) \frac{P(y|x)}{P'(y|x)} \tag{38}$$

$$\approx \arg\max_{h \in \mathcal{H}} \sum_{y \in \mathcal{Y}} f_\theta(h, y) \frac{P(y|x)}{P'(y|x)}, \quad y_1, \cdots, y_{|\mathcal{Y}|} \sim P'(y|x) \tag{39}$$

$\square$

Apart from the fact that even precisely calculating $P'(y|x)$ is also difficult, we can induce the following theorem:

**Theorem 6.** *When $|\mathcal{Y}|$ is enough large to satisfy the law of large numbers and $P'(y|x)$ equals a discrete uniform distribution $\mathcal{U}(0, |\mathcal{Y}|)$, Equation 39 is equivalent to Model-based MBR (MBMBR) of Jinnai et al. (2024b).*

*Proof.*

$$\arg\max_{h \in \mathcal{H}} \sum_{y \in \mathcal{Y}} f_\theta(h, y) \frac{P(y|x)}{P'(y|x)}, \quad y_1, \cdots, y_{|\mathcal{Y}|} \sim P'(y|x) \tag{40}$$

$$= \arg\max_{h \in \mathcal{H}} \frac{1}{|\mathcal{Y}|} \sum_{y \in \mathcal{Y}} f_\theta(h, y) P(y|x), \quad y_1, \cdots, y_{|\mathcal{Y}|} \sim \mathcal{U}(0, |\mathcal{Y}|) \tag{41}$$

$$= \arg\max_{h \in \mathcal{H}} \sum_{y \in \mathcal{Y}} f_\theta(h, y) P(y|x), \quad y_1, \cdots, y_{|\mathcal{Y}|} \sim \mathcal{U}(0, |\mathcal{Y}|) \tag{42}$$

$\square$

From Theorem 6, we can understand that MBMBR is an effective approach when sampling methods are unreliable. Based on the interpretation from the viewpoint of BOC, Equation 42 estimates the importance for each $y$ through prior $P(y|x)$, which can be used for pruning $y$.

Even though our interpretation can explain the pruning of pseudo-references based on priors in BOC, pruning hypotheses are out-of-scope of this interpretation.

## D    CORRELATION OF BIAS AND DIVERSITY TO PERFORMANCE

We further investigate whether our analysis in §5.2 is consistent when metrics used in MBR decoding and performance evaluation are different.

**Settings**    Based on the inherited settings from §5.2, we changed the performance evaluation metrics, COMET and BERTScore to BLEURT (Sellam et al., 2020) and chrF++ (Popović, 2015; 2017). We used BLEURT on single sentence generation tasks, WMT19 En-De and En-Ru, XSum, MSCOCO, and NoCaps. Since SAMSum is a multiple-sentence generation task and BLEURT cannot handle it, we used chrF++ instead.

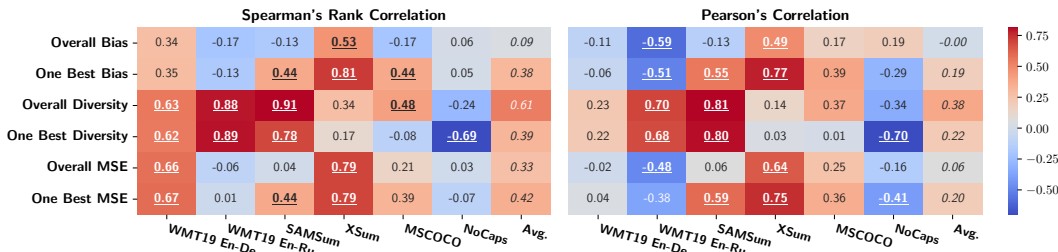

Figure 3: Correlation between measures in our decomposition and performance for each dataset when using different metrics in decoding and performance evaluation. The notations are the same as Figure 1.

**Results** Figure 3 shows the correlation. Similar to the results in §5.2, the measures, i.e., ONE BEST BIAS, OVERALL DIVERSITY, and ONE BEST MSE in Spearman's rank correlation correlate well with the rank in performance, even though these correlation values are degraded by different evaluation metrics from decoding time. The lower correlation values in Pearson's correlation than Spearman's rank correlation also show similar tendencies in §5.2 and indicate the difficulty of precisely estimating the performance values from these measures. From these results, we can confirm that correlation tendencies are consistent when changing the performance evaluation metrics.

# E  BIAS AND DIVERSITY TRADE-OFF

Similar to Appendix D, we further investigate whether our analysis in §5.3 is consistent when metrics used in MBR decoding and performance evaluation are different.

**Settings** We inherited the setting of Appendix D. Thus, COMET and BERTScore used in MBR decoding are replaced with BLEURT and chrF++ in performance evaluation.

**Results** Figure 4 shows the results. We can see the changed performances in the subfigures of the rightmost column. The entire tendencies of beam decoding are almost the same as Figure 2, excluding the case of the performance drop in SAMSum, whose evaluation metric is changed from BERTScore to chrF++. However, this behavior is reasonable considering the highest `One Best Bias` and lowest `Overall Diversity` of beam decoding in SAMSum. This result shows the possibility of adopting bias and diversity in a metric to the estimation of performance in other evaluation metrics. On the other hand, these relationships are not always consistent as represented in the uncorrelated values on NoCaps that permit diversified generation by its 10 gold references.

# F  BIAS AND DIVERSITY OF MAMBR

Figures 5, 6, and 7 show the bias and diversity corresponding to the results in Tables 1, 2, and 3, respectively. The results show that MAMBR actually increases the diversities in WMT19 En-De and En-Ru and SAMSum but not in the other dataset. Thus, this improvement depends on the datasets. On the other hand, we can see the improvement of bias in some cases. This is reasonable because using multiple metric models itself is an ensembling approach and can contribute to performance improvement.

# G  EXPERIMENTAL RESULTS ON THE FIRST 1000 EXAMPLES

To consider more detailed configurations and reveal the possibility of more efficient investigation, we conducted additional evaluation using only the first 1000 examples for each dataset based on the setting of Jinnai et al. (2024b).

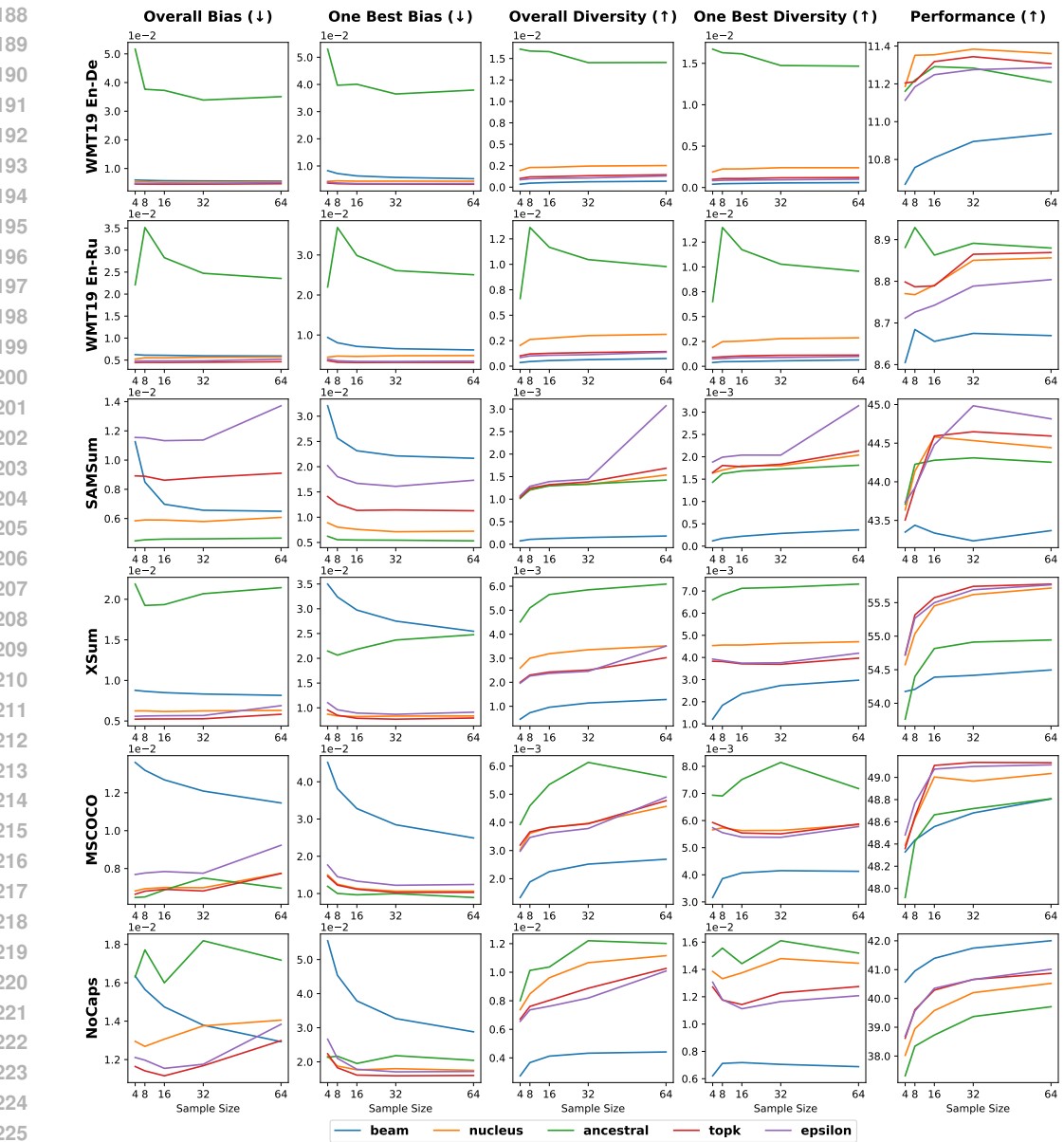

Figure 4: The relationship between bias, diversity, and performance in MBR decoding when using different metrics in decoding and performance evaluation. The notations are the same as Figure 2.

## G.1 HYPOTHESES GENERATED BY DIFFERENT SAMPLING STRATEGIES

Figures 8 to 12 present the bias and diversity decomposition plots for different hypothesis generation strategies. The results indicate that differences in the generated hypotheses influence performance in some cases, whereas the overall tendencies of the sampling strategy used for generating pseudo-references remain similar despite these variations.

## G.2 MAMBR

Tables 4 to 6 show the MAMBR results for the first 1000 lines. From these results, we observe a similar trend to those obtained when the dataset is fully used, as described in §5.4. Similarly, Figures 13 to 15 demonstrate that the results are nearly identical to those obtained when the dataset is fully utilized.

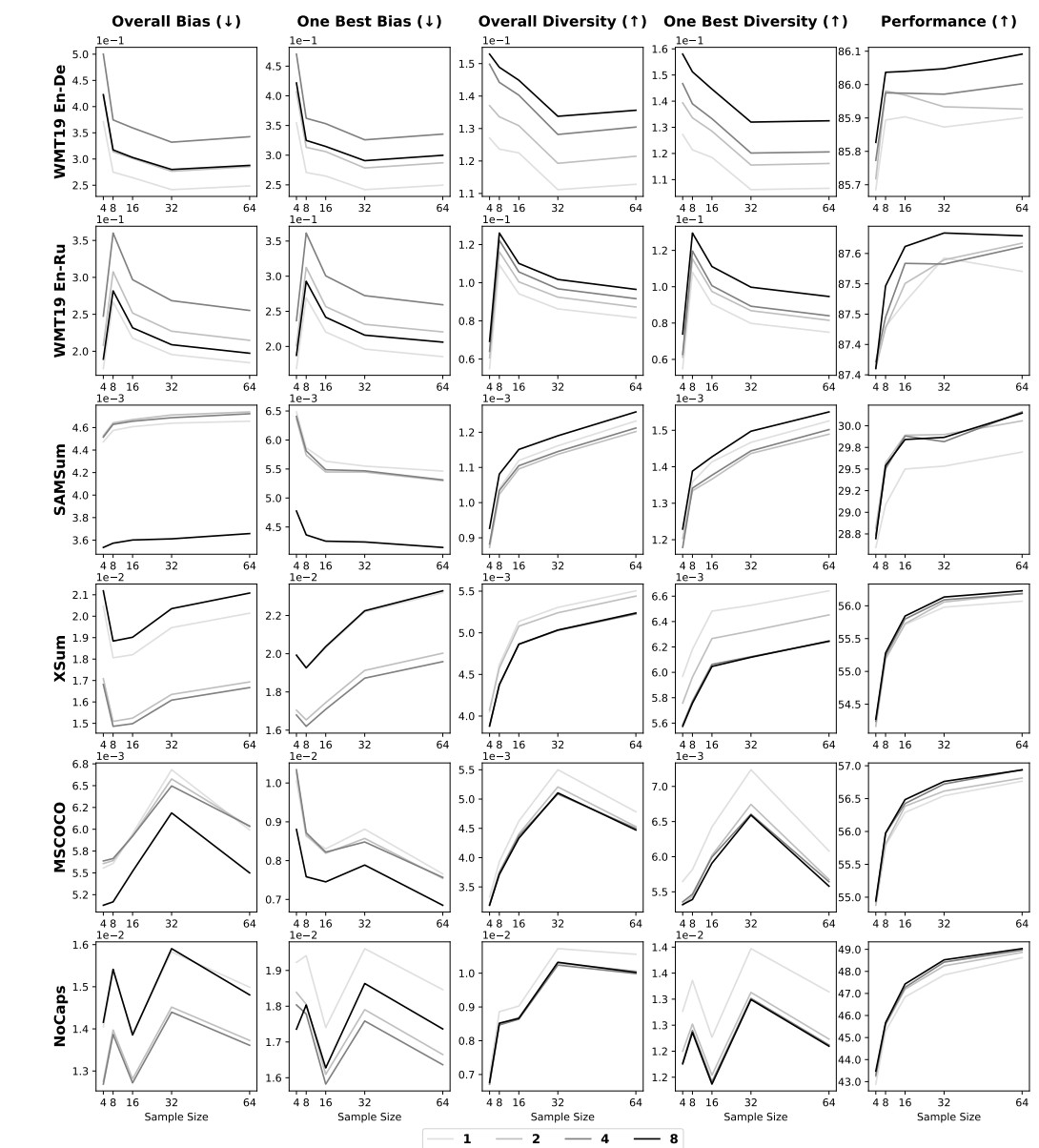

Figure 5: The relationship between bias, diversity, and performance in MAMBR decoding with pseudo-references generated by ancestral sampling. The notations are the same as Figure 6.

## H LIMITATION

Although our bias-diversity decomposition for MBR decoding can explain the behavior of pseudo-references and utility functions, a theoretical explanation for the effectiveness of the used hypotheses is a model-side behavior and thus beyond the scope of our analysis. Therefore, corresponding to this limitation, we conduct a limited empirical analysis presented in Appendix G.1, similar to previous works (Eikema & Aziz, 2020; Fernandes et al., 2022; Freitag et al., 2023a).

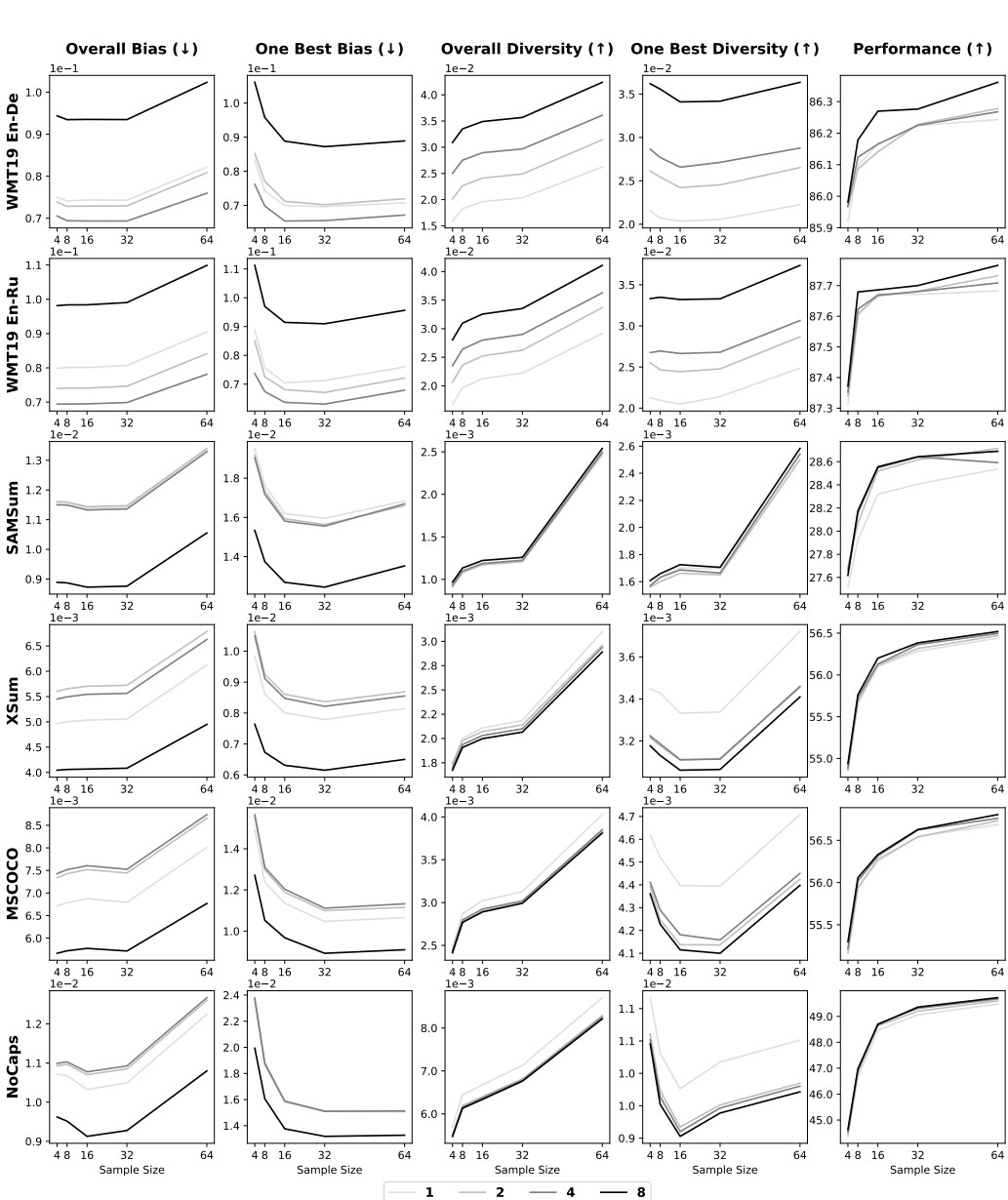

Figure 6: The relationship between bias, diversity, and performance in MAMBR decoding with pseudo-references generated by epsilon sampling. The lines indicate the score for each number of used metric models. Other notations are the same as Figure 6.

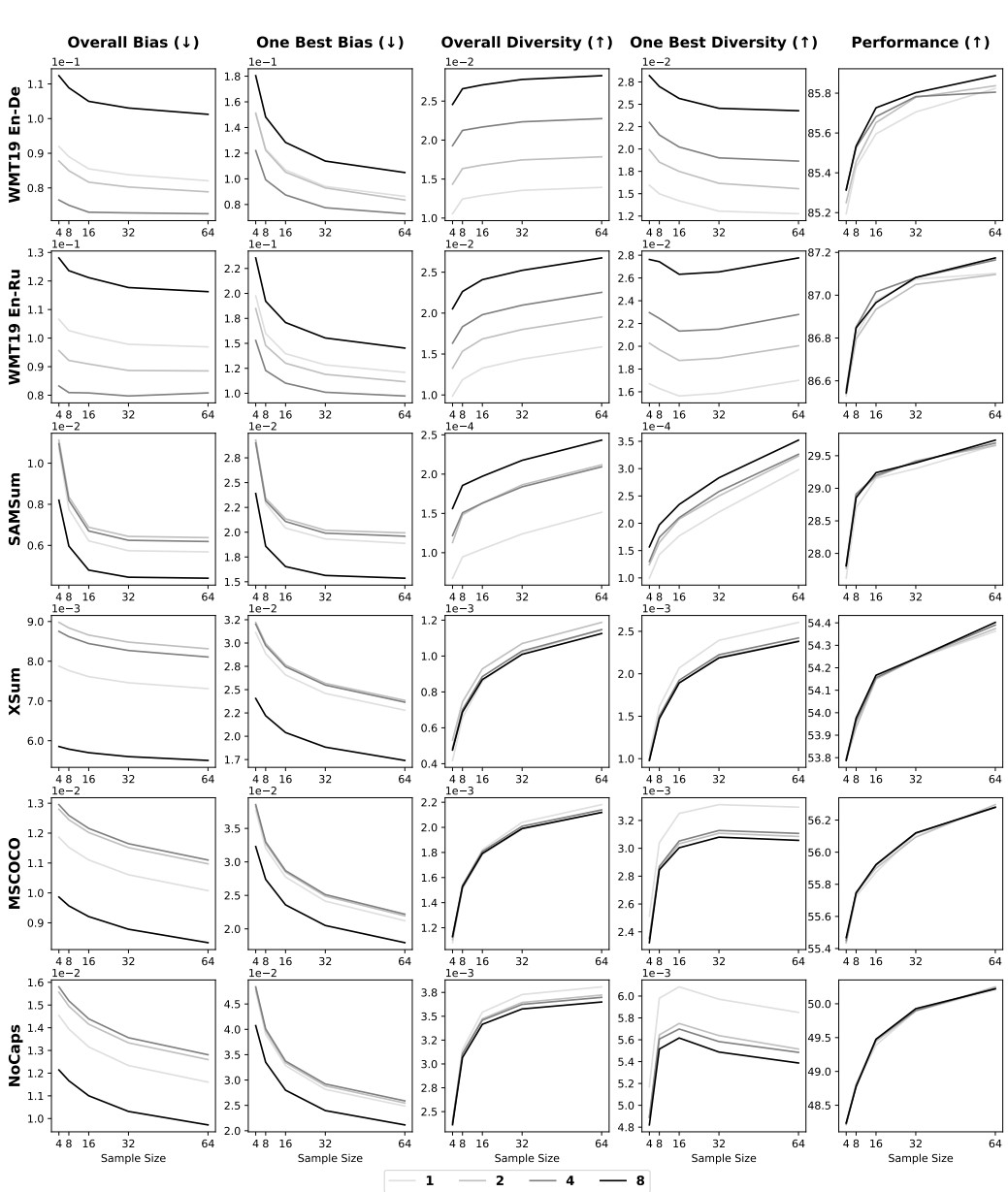

Figure 7: The relationship between bias, diversity, and performance in MAMBR decoding with pseudo-references generated by beam decoding. The notations are the same as Figure 6.

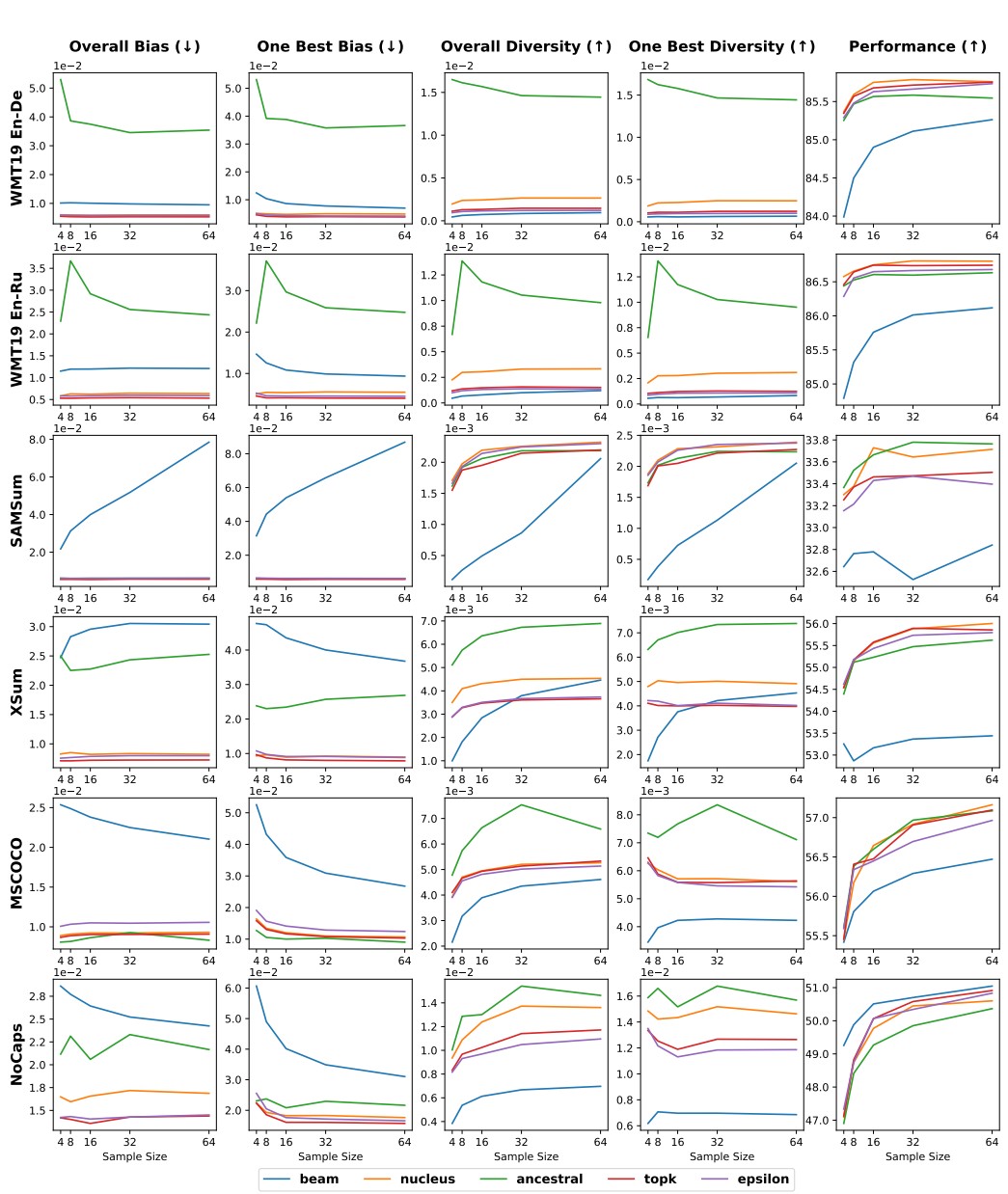

Figure 8: The relationship between bias, diversity, and performance on the first 1000 lines of each dataset in MBR decoding with hypotheses generated by beam decoding. The notations are the same as Figure 2.

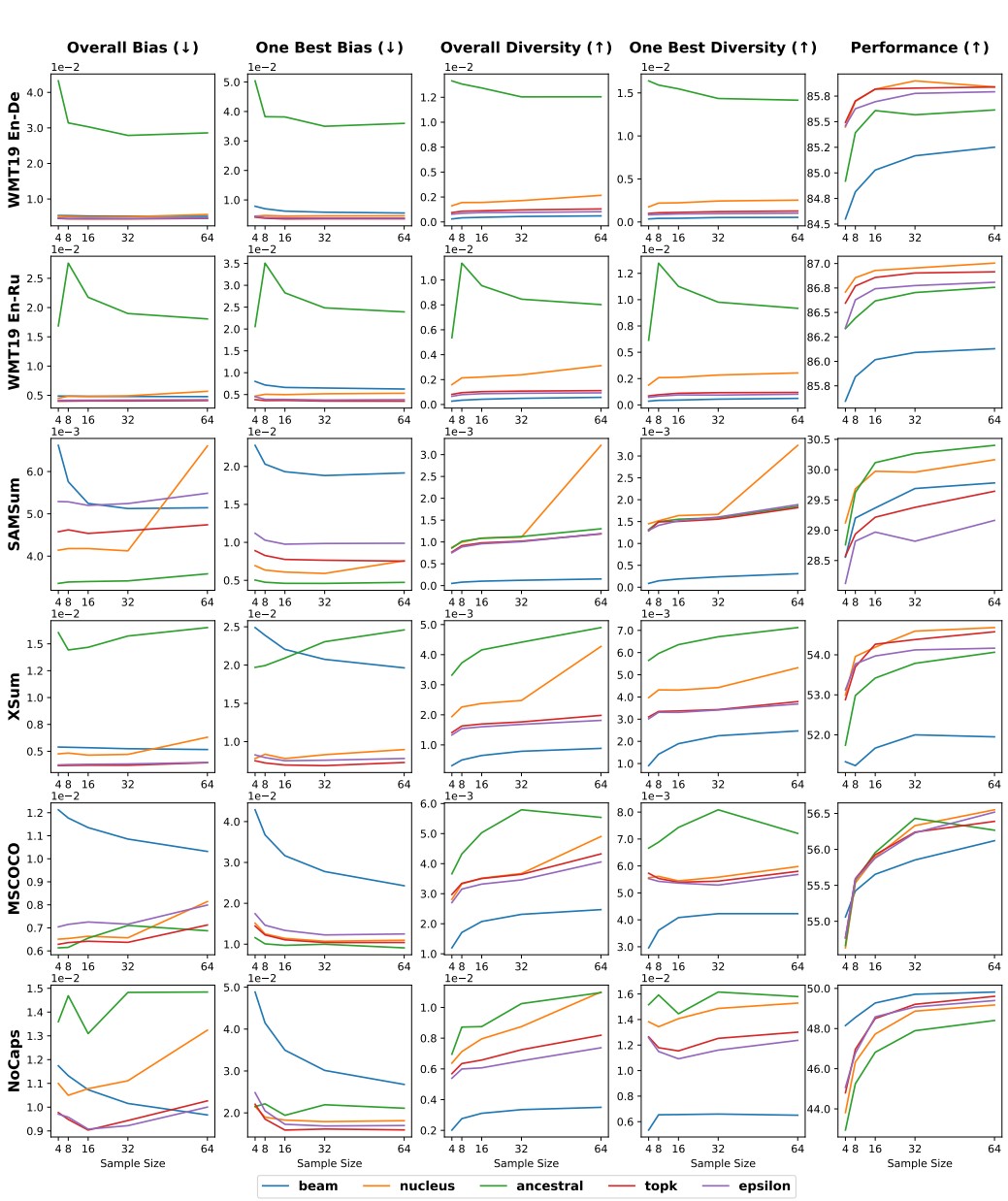

Figure 9: The relationship between bias, diversity, and performance on the first 1000 lines of each dataset in MBR decoding with hypotheses generated by nucleus sampling. The notations are the same as Figure 2.

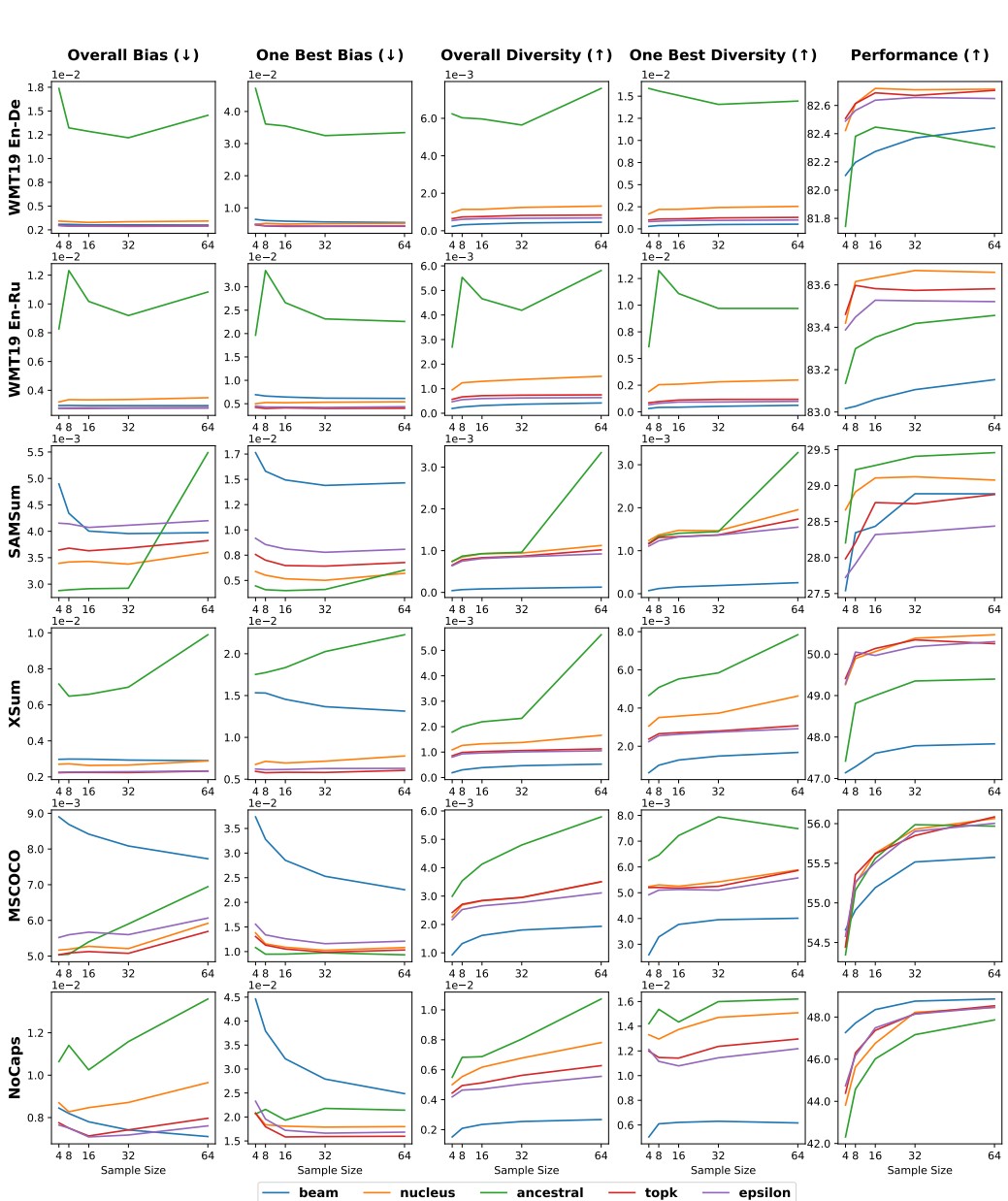

Figure 10: The relationship between bias, diversity, and performance on the first 1000 lines of each dataset in MBR decoding with hypotheses generated by ancestral sampling. The notations are the same as Figure 2.

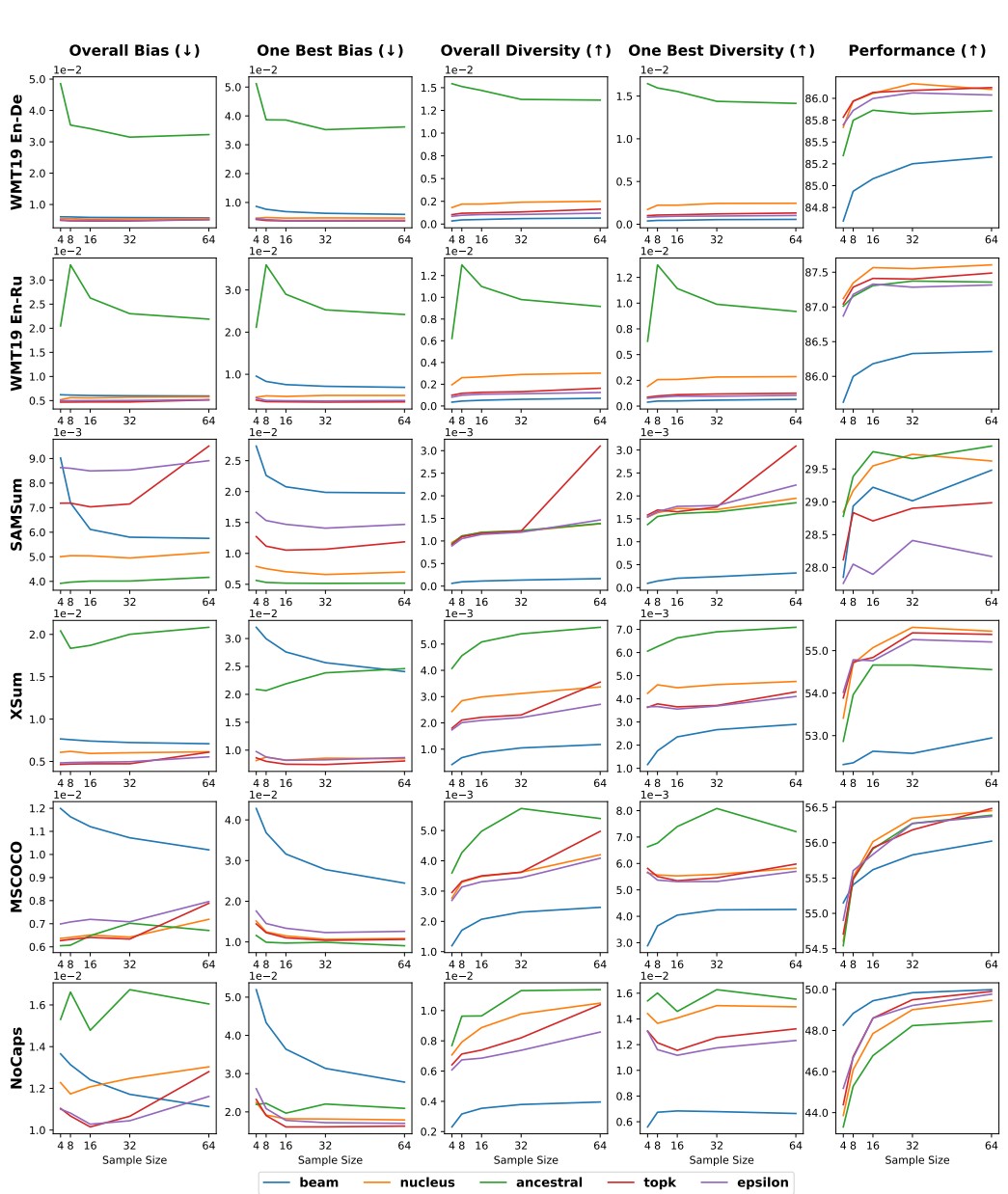

Figure 11: The relationship between bias, diversity, and performance on the first 1000 lines of each dataset in MBR decoding with hypotheses generated by top-k sampling. The notations are the same as Figure 2.

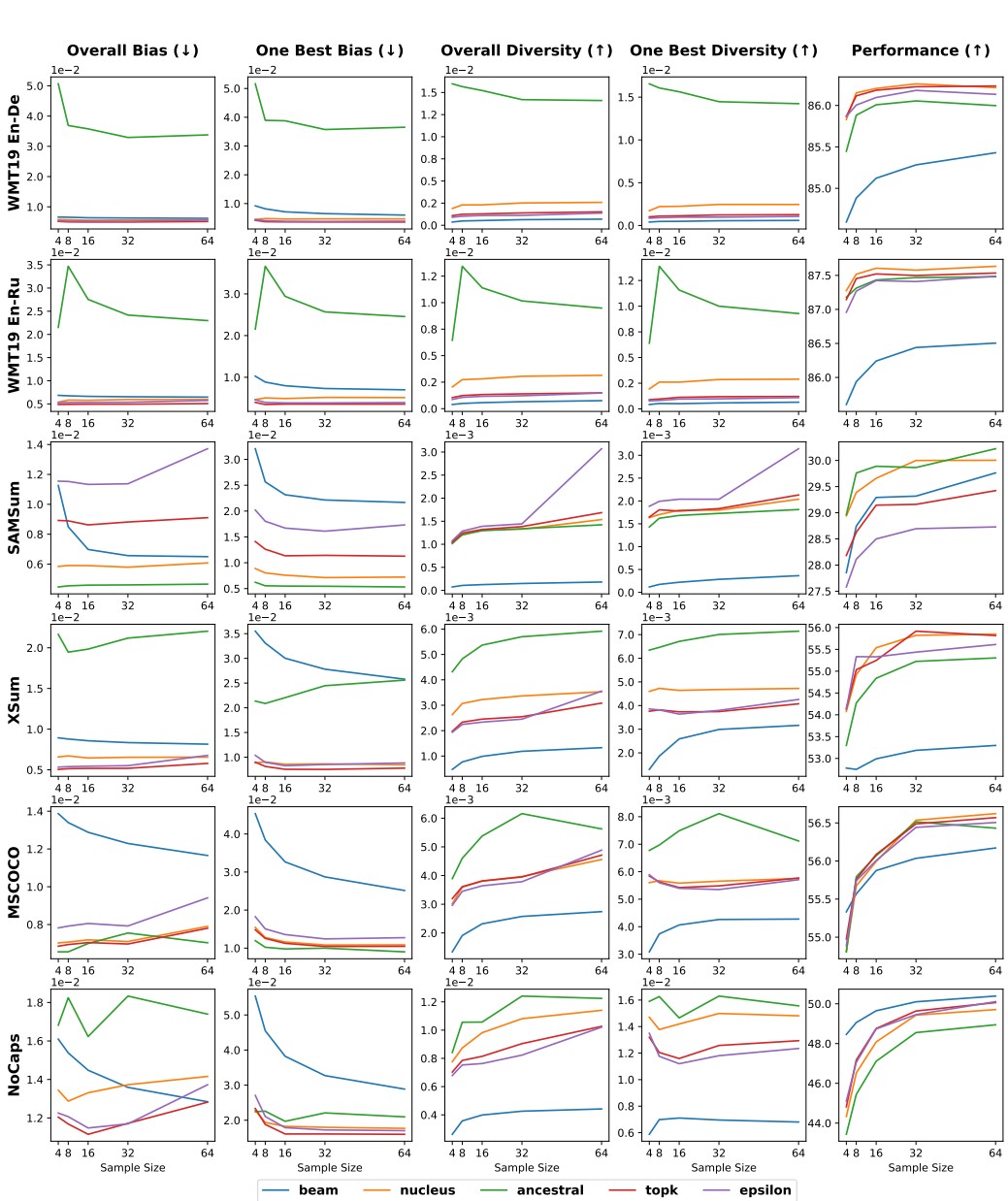

Figure 12: The relationship between bias, diversity, and performance on the first 1000 lines of each dataset in MBR decoding with hypotheses generated by epsilon sampling. The notations are the same as Figure 2.

Table 4: Results of MAMBR with samples generated by ancestral sampling. Notations are the same as Table 1.

| | | WMT19 En-De | | | | | WMT19 En-Ru | | | | |
|---|---|---|---|---|---|---|---|---|---|---|---|
| Num. of Samples | | 4 | 8 | 16 | 32 | 64 | 4 | 8 | 16 | 32 | 64 |
| | 1 | 84.5 | 84.7 | 84.8 | 85.0 | 85.1 | **85.8** | **86.1** | 86.3 | **86.5** | 86.5 |
| Num. of Models | 2 | 84.5 | **84.8** | **85.0** | **85.1** | 85.2 | 85.8 | 86.1 | 86.3 | 86.4 | 86.5 |
| | 4 | 84.6 | **84.8** | **85.0** | **85.1** | 85.2 | 85.8 | 86.1 | **86.4** | 86.5 | 86.6 |
| | 8 | **84.7** | **84.8** | **85.0** | **85.1** | **85.3** | 85.8 | 86.1 | 86.3 | 86.5 | 86.6 |
| | | SAMSum | | | | | XSum | | | | |
| Num. of Samples | | 4 | 8 | 16 | 32 | 64 | 4 | 8 | 16 | 32 | 64 |
| | 1 | 28.6 | 29.1 | 29.5 | 29.5 | 29.7 | 53.3 | 54.1 | **55.0** | 55.1 | 55.2 |
| Num. of Models | 2 | **28.8** | **29.6** | **29.9** | **29.9** | 30.1 | 53.2 | 54.2 | **55.0** | **55.3** | **55.3** |
| | 4 | 28.7 | 29.5 | **29.9** | 29.8 | **30.2** | 53.3 | 54.2 | **55.0** | **55.3** | **55.3** |
| | 8 | 28.7 | 29.5 | 29.8 | **29.9** | 30.1 | **53.4** | **54.3** | **55.0** | 55.2 | **55.3** |
| | | MSCOCO | | | | | NoCaps | | | | |
| Num. of Samples | | 4 | 8 | 16 | 32 | 64 | 4 | 8 | 16 | 32 | 64 |
| | 1 | 54.8 | 55.8 | 56.1 | 56.4 | 56.6 | 43.2 | 45.3 | 46.9 | 48.2 | 49.0 |
| Num. of Models | 2 | 54.8 | 55.8 | 56.2 | 56.4 | 56.6 | 43.4 | 45.7 | 47.2 | 48.7 | 49.1 |
| | 4 | 54.8 | **55.9** | **56.4** | 56.5 | **56.8** | 43.8 | 45.8 | **47.5** | 48.8 | **49.5** |
| | 8 | **54.9** | **55.9** | 56.3 | **56.6** | **56.8** | **43.9** | **45.9** | 47.4 | **49.0** | **49.5** |

Table 5: Results of MAMBR with samples generated by epsilon sampling. Notations are the same as Table 1.

| | | WMT19 En-De | | | | | WMT19 En-Ru | | | | |
|---|---|---|---|---|---|---|---|---|---|---|---|
| Num. of Samples | | 4 | 8 | 16 | 32 | 64 | 4 | 8 | 16 | 32 | 64 |
| | 1 | 85.2 | 85.4 | 85.6 | 85.6 | 85.6 | **86.7** | 87.0 | **87.1** | 87.0 | 87.1 |
| Num. of Models | 2 | **85.3** | 85.5 | 85.6 | **85.7** | 85.7 | **86.7** | 87.0 | **87.1** | **87.1** | 87.1 |
| | 4 | **85.3** | 85.5 | 85.6 | **85.7** | 85.7 | **86.7** | 87.0 | **87.1** | 87.0 | 87.1 |
| | 8 | **85.3** | **85.6** | **85.7** | **85.7** | **85.8** | **86.7** | **87.1** | **87.1** | 87.0 | **87.2** |
| | | SAMSum | | | | | XSum | | | | |
| Num. of Samples | | 4 | 8 | 16 | 32 | 64 | 4 | 8 | 16 | 32 | 64 |
| | 1 | 27.5 | 27.9 | 28.3 | 28.4 | 28.5 | 54.1 | 55.1 | 55.1 | 55.4 | 55.5 |
| Num. of Models | 2 | **27.7** | 28.1 | 28.5 | **28.6** | **28.7** | 54.1 | **55.2** | 55.1 | 55.4 | 55.4 |
| | 4 | **27.7** | **28.2** | 28.5 | **28.6** | 28.6 | 54.1 | **55.2** | 55.2 | **55.5** | **55.6** |
| | 8 | 27.6 | **28.2** | **28.6** | **28.6** | **28.7** | **54.2** | **55.2** | **55.3** | 55.4 | **55.6** |
| | | MSCOCO | | | | | NoCaps | | | | |
| Num. of Samples | | 4 | 8 | 16 | 32 | 64 | 4 | 8 | 16 | 32 | 64 |
| | 1 | 55.0 | 55.8 | 56.0 | 56.4 | 56.6 | 45.0 | 46.9 | 48.6 | 49.5 | 49.8 |
| Num. of Models | 2 | 55.0 | 55.7 | 56.1 | 56.4 | 56.6 | 45.0 | 47.1 | 48.9 | 49.5 | 50.1 |
| | 4 | 55.0 | **55.9** | **56.2** | **56.6** | 56.7 | **45.2** | **47.2** | 48.9 | **49.8** | 50.2 |
| | 8 | **55.1** | 55.8 | **56.2** | 56.5 | **56.8** | **45.2** | **47.2** | **49.0** | 49.8 | 50.3 |

Table 6: Results of MAMBR with samples generated by beam decoding. Notations are the same as Table 1.

|  |  | WMT19 En-De | | | | | WMT19 En-Ru | | | | |
| --- | --- | --- | --- | --- | --- | --- | --- | --- | --- | --- | --- |
| Num. of Samples | | 4 | 8 | 16 | 32 | 64 | 4 | 8 | 16 | 32 | 64 |
| Num. of Models | 1 | 85.1 | 85.3 | 85.4 | 85.4 | 85.3 | **86.7** | **86.9** | **86.9** | **86.9** | 86.9 |
| | 2 | 85.1 | 85.4 | **85.5** | 85.4 | 85.4 | **86.7** | 86.8 | **86.9** | **86.9** | **87.0** |
| | 4 | **85.2** | 85.4 | **85.5** | 85.4 | 85.4 | **86.7** | 86.8 | **86.9** | **86.9** | 86.9 |
| | 8 | **85.2** | **85.5** | **85.5** | **85.5** | **85.5** | 86.6 | **86.9** | **86.9** | **86.9** | 86.9 |
|  |  | SAMSum | | | | | XSum | | | | |
| Num. of Samples | | 4 | 8 | 16 | 32 | 64 | 4 | 8 | 16 | 32 | 64 |
| Num. of Models | 1 | 27.6 | 28.7 | **29.2** | 29.3 | **29.7** | **52.8** | 52.8 | **53.1** | **53.2** | 53.3 |
| | 2 | 27.8 | 28.9 | **29.2** | 29.4 | **29.7** | 52.7 | **52.9** | 53.0 | **53.2** | 53.4 |
| | 4 | 27.8 | 28.9 | **29.2** | 29.4 | **29.7** | **52.8** | 52.8 | 53.0 | **53.2** | 53.3 |
| | 8 | 27.8 | 28.9 | **29.2** | 29.4 | **29.7** | 52.7 | **52.9** | **53.1** | **53.2** | 53.3 |
|  |  | MSCOCO | | | | | NoCaps | | | | |
| Num. of Samples | | 4 | 8 | 16 | 32 | 64 | 4 | 8 | 16 | 32 | 64 |
| Num. of Models | 1 | **55.3** | **55.5** | **55.9** | 56.0 | 56.2 | **48.5** | 49.0 | 49.6 | 50.1 | **50.5** |
| | 2 | **55.3** | **55.5** | 55.8 | **56.1** | 56.2 | **48.5** | 49.0 | **49.8** | 50.2 | **50.5** |
| | 4 | **55.3** | **55.5** | 55.8 | 56.0 | **56.3** | **48.5** | 49.1 | 49.7 | 50.2 | **50.5** |
| | 8 | **55.3** | **55.5** | 55.8 | **56.1** | 56.2 | **48.5** | **49.2** | 49.7 | **50.3** | **50.5** |

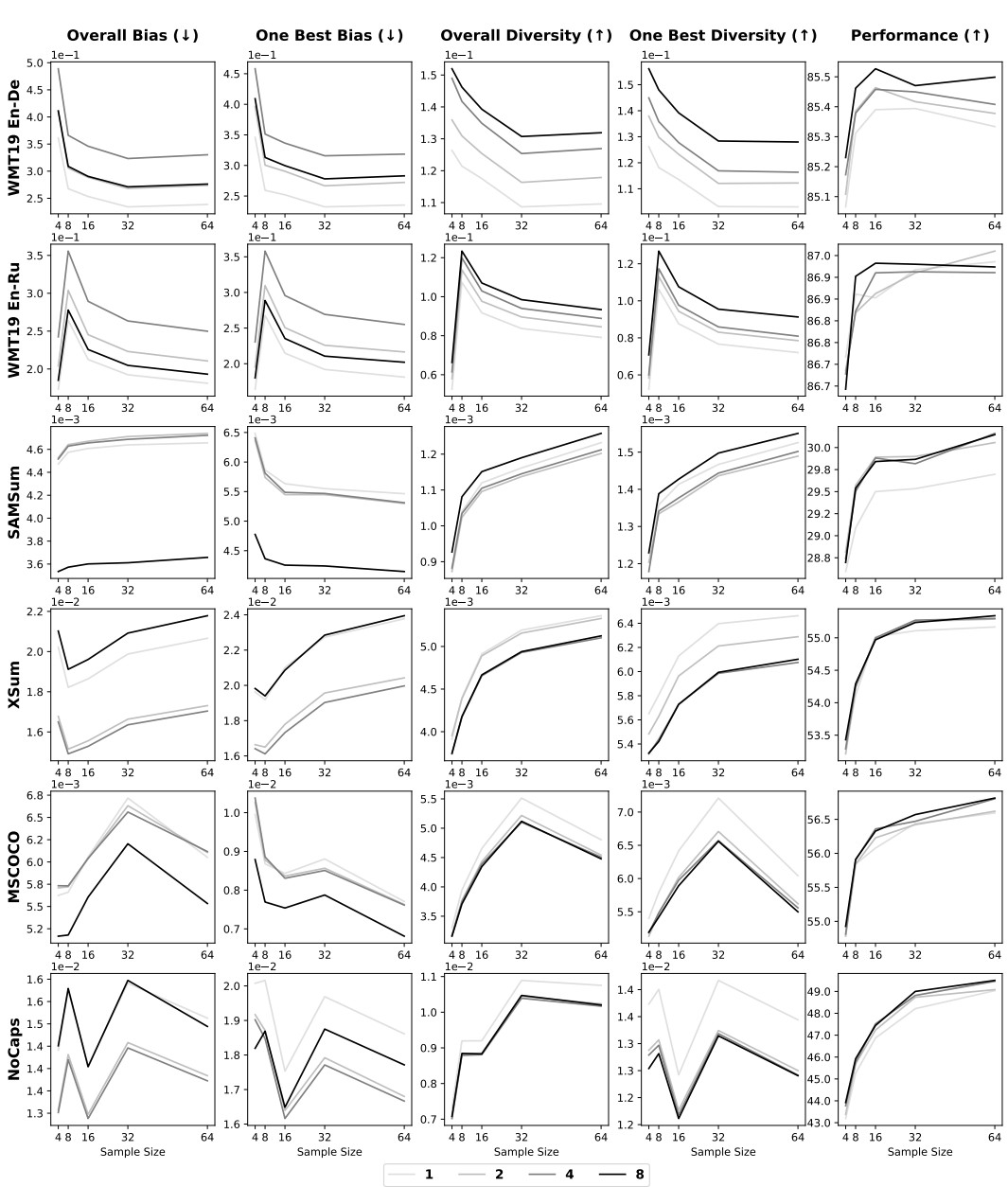

Figure 13: The relationship between bias, diversity, and performance on the first 1000 lines of each dataset in MBR decoding with pseudo-references generated by ancestral sampling. The notations are the same as Figure 6.

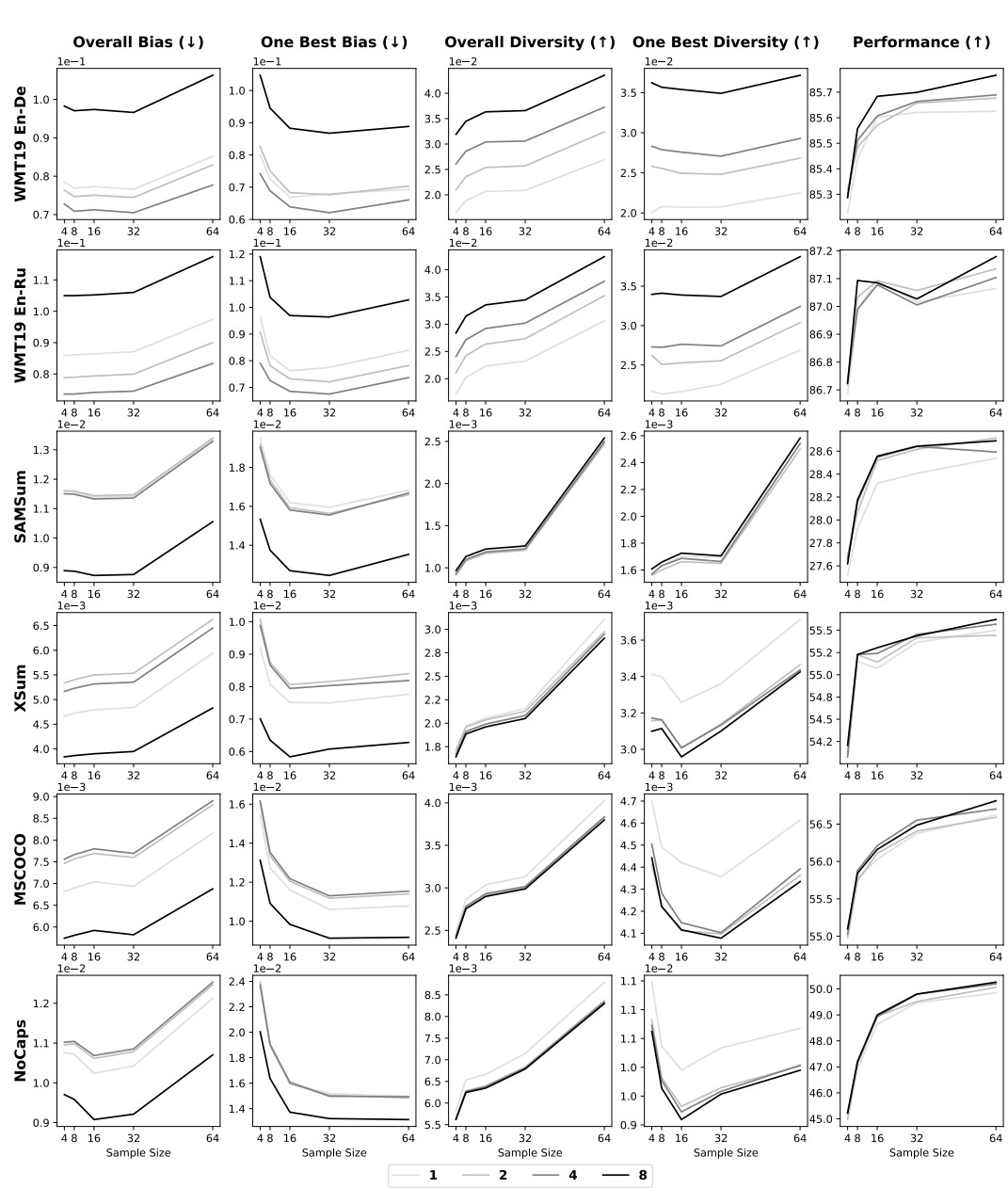

Figure 14: The relationship between bias, diversity, and performance on the first 1000 lines of each dataset in MBR decoding with pseudo-references generated by epsilon sampling. The notations are the same as Figure 6.

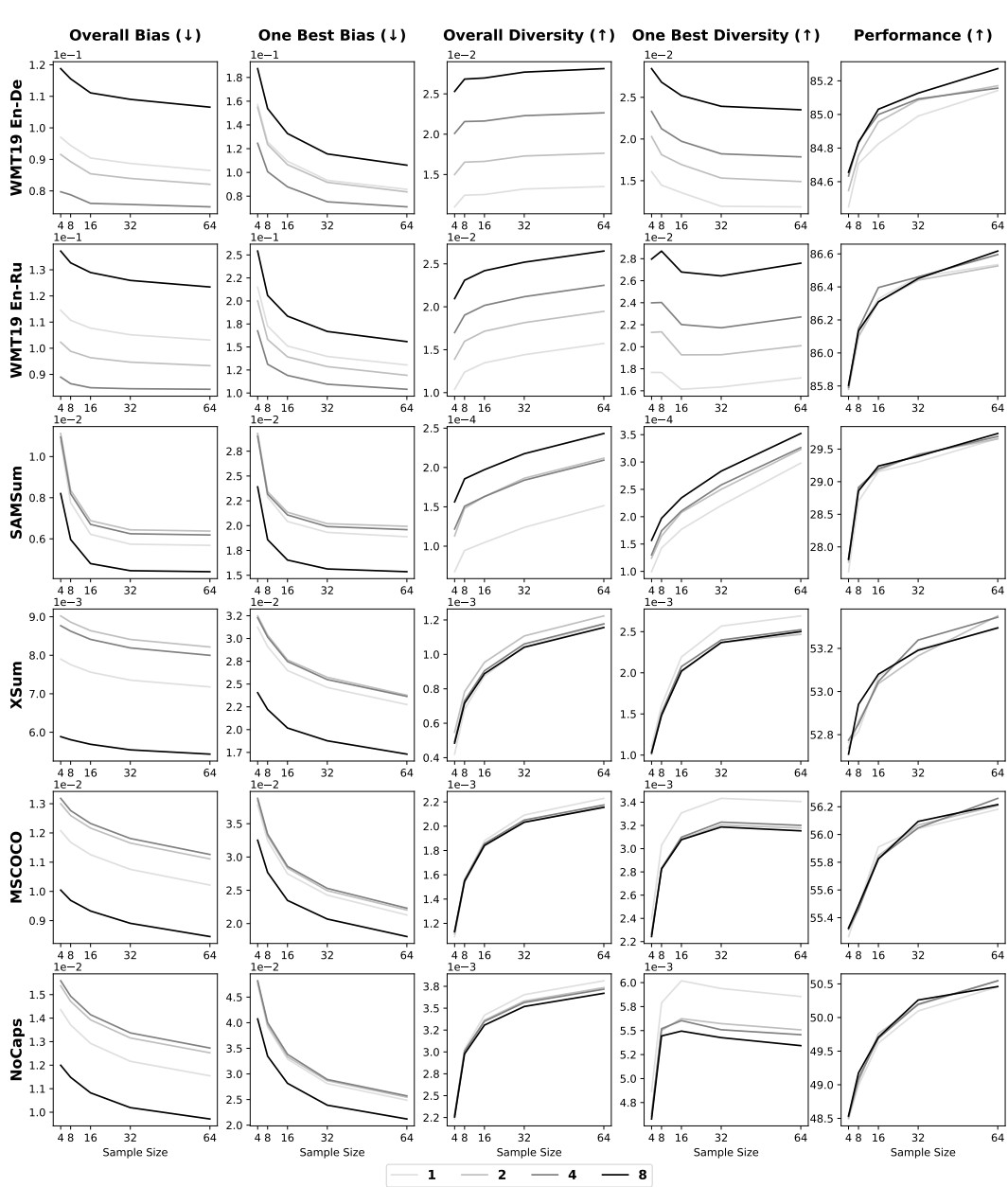

Figure 15: The relationship between bias, diversity, and performance on the first 1000 lines of each dataset in MBR decoding with pseudo-references generated by beam decoding. The notations are the same as Figure 6.

