# OpenReview forum: "Theoretical Aspects of Bias and Diversity in Minimum Bayes Risk Decoding"
_ICLR.cc/2025/Conference — Submitted to ICLR 2025_

### Official Review · Reviewer_Fq8u · 2024-11-03

**Soundness:** 3
**Presentation:** 2
**Contribution:** 3
**Rating:** 5
**Confidence:** 2

**Summary:**

The paper offers a theoretical perspective on MBR decoding by decomposing the discrepancy between automatic MBR scores with the human evaluations using a bias-diversity decomposition. Their analysis suggest that one may want systems that minimize bias but also the diversity should be increased, which aligns with previous paper’s empirical results. They then analyse performance of MBR decoding in several scenarios and show that the analysis holds, where models with higher diversity and lower bias have better correlations. They then introduce MAMBR, which aims to maximise diversity by modifying the behavior of the utility functions. Results show marginal gains in performance across different tasks, while using a similar number of sampled pseudo-references.

**Strengths:**

- The paper decomposes the MBR decoding into cleaner interpretations of bias and diversity, which can shed light on the MBR expression.  The resulting expression appears fairly elegant, especially if it explains existing empirical observations, which the paper linked to, where increasing diversity was shown to be helpful.

- Their results show that increasing the diversity through MAMBR can result in better performance across 3 different tasks and various numbers of models without the need for any further references.

**Weaknesses:**

- As I understand it, the approach uses pseudo-references, which are then used to approximate the human evaluation scores (by averaging over different references), which may impact the results. The generated hypothesis and the references appear to be generated from the same model just using different sampling strategies, which may result on a lot of similarity between the two systems and underestimate the bias for different approaches in Figure 2 than if independent human evlauation scores were used to assess the bias.

- The results in Table 1, 2 and 3 seem quite marginal, and there hasn’t been any quantification of the significance of the results.

**Questions:**

- I personally found parts of the experimental section a bit confusing. Though the theoretical aspect was mostly quite clear and made sense, from section 5 onwards, I found the results a bit less clear, possibly due to my having less familiarity with previous works. I wasn’t clear on what was used as the references, the hypothesis and how performance was measured. As I understand it, the same models generated both the hypothesis and the references which are used in MBR decoding (just with different sampling approaches). Is this correct? Also are there are ground-truth human evaluation scores available, which performance is measured against, or are all results pseudo results using equation 10?

- In MAMBR you stated that “we train evaluation metrics with different initial random seeds to generate Θ as a set of diverse model parameters” but the evaluation metrics used appear to be external packages (e.g. COMET, BERTScore) that were leveraged. How exactly did you then implement MAMBR?

- In line 93 you say “Here, instead of using …” Is the here intentional, and describing the previous equation, or just introducing the new approach of using manual selection of the best hypothesis (equation 3). There are a few more instances of this in the earlier sections

---

> ### Author Response · Authors · 2024-12-01
> **Responses to Weaknesses and Questions**
>
> Thank you for reviewing our paper and sharing your feedback. We have updated our manuscript based on your opinion.
>
> ## Responses to Weaknesses
>
> > As I understand it, the approach uses pseudo-references, which are then used to approximate the human evaluation scores (by averaging over different references), which may impact the results. The generated hypothesis and the references appear to be generated from the same model just using different sampling strategies, which may result on a lot of similarity between the two systems and underestimate the bias for different approaches in Figure 2 than if independent human evaluation scores were used to assess the bias.
>
> Basically, MBR decoding uses the hypothesis and pseudo-references generated by the same model. Thus, we follow this setting to investigate the behavior of MBR decoding in common situations. Furthermore, using more than two models is a so-called model combination and totally out-of-scope of our paper. Therefore, we don't refer to this setting in our paper. However, to include the limitation of MBR decoding, we explain the necessity of investigating the use of hypothesis in the limitation of the Appendix part.
>
> > The results in Table 1, 2 and 3 seem quite marginal, and there hasn’t been any quantification of the significance of the results.
>
> As explained in subsection 3.3.2, we aim to show the exchangeability of the diversity of pseudo references and utility functions' output. Therefore, if there are no changes between the results of MAMBR and those of diversified pseudo references, this supports our theoretical analysis and shows the success of our experimental results.
>
> ## Responses to Questions
>
> > I personally found parts of the experimental section a bit confusing. Though the theoretical aspect was mostly quite clear and made sense, from section 5 onwards, I found the results a bit less clear, possibly due to my having less familiarity with previous works. I wasn’t clear on what was used as the references, the hypothesis and how performance was measured. As I understand it, the same models generated both the hypothesis and the references which are used in MBR decoding (just with different sampling approaches). Is this correct? Also are there are ground-truth human evaluation scores available, which performance is measured against, or are all results pseudo results using equation 10?
>
> As explained in section 5.1, we used the commonly used datasets in Machine Translation, Text Summarization, and Image Captioning to evaluate the performance of MBR decoding. Thus, we used their test split for the evaluation using the reference-based metrics COMET and BERTScore. Also, we used the same specific models for each task to generate both hypothesis and pseudo references as the orthodontically used setting in MBR decoding. Regarding the correlation to human evaluation results of COMET and BERTScore, we have added explanations on lines 319 to 322. As explained in section 5.2, Equation 10 is used to calculate only overall and one-best bias in Figures 1 and 2.
>
> > In MAMBR you stated that “we train evaluation metrics with different initial random seeds to generate Θ as a set of diverse model parameters” but the evaluation metrics used appear to be external packages (e.g. COMET, BERTScore) that were leveraged. How exactly did you then implement MAMBR?
>
> We preserved multiple checkpoints locally and then stored them onto GPU memory when running MBR decoding. For that purpose, we modified the original implementation of BERTScore, whereas COMET does not require modifications. We will release our code when the paper is accepted.
>
> > In line 93 you say “Here, instead of using …” Is the here intentional, and describing the previous equation, or just introducing the new approach of using manual selection of the best hypothesis (equation 3). There are a few more instances of this in the earlier sections
>
> Thank you so much for pointing out the need for the details. To support this explanation, we have added citations of conventional approaches, assuming that human judgment results are the ideal decision in MBR decoding.

---

> > ### Author Response · Authors · 2024-12-04
> > **Reminder**
> >
> > We believe that the reason you are not responding is not because you irresponsibly abandoned your role as a reviewer but because the concerns regarding this paper have been resolved. Therefore, if the concerns have indeed been resolved, we kindly request you to update your score regarding Soundness, Presentation, Contribution, and Overall Rating to reflect the outcome that no concerns remain, as is typically expected in the role of a reviewer after this discussion period.

---

### Official Review · Reviewer_zsMt · 2024-11-04

**Soundness:** 2
**Presentation:** 1
**Contribution:** 1
**Rating:** 3
**Confidence:** 3

**Summary:**

This paper introduces a bias-diversity decomposition for Minimum Bayes Risk Decoding. The bias reflects the distance between the utility functions and human evaluations, while diversity represents the variation. The analysis and experiments show that a lower bias and a higher diversity will lead to better model performance. It also proposes a Metric-augmented MBR, which use multiple utility functions with various parameters to enhance diversity.

**Strengths:**

1. It provides a bias-diversity view of MBR, which highlights the importance of several factors in MBR, such as the quality of utility functions / pseudo reference.
2. The analysis is verified by the correlation between the bias / diversify of MBR with the model performance. It further investigates the influence of different sampling methods and the size of pseudo preferences.

**Weaknesses:**

1. The main weakness is the novelty and the contribution is limited. Although it proposes a bias-diversity decomposition of MBR, such decomposition is straightforward. The observations listed in Section 3.3 have mostly been covered in other literature, resulting in limited new insights.
2. The Performance of MAMBR with ancestral sampling/epsilon sampling/beam decoding is not significantly better than the baseline model, which weakens the paper's claim.

**Questions:**

For all three tasks, the utility function and the evaluation metric are the same. What will Figure 1 be if the utility function is different from the evaluation metric?

---

> ### Author Response · Authors · 2024-12-01
> **Responses to Weaknesses and Questions**
>
> Thank you for reviewing our paper and sharing your feedback. We have updated our manuscript based on your opinion.
>
> ## Responses to Weaknesses
>
> > The main weakness is the novelty and the contribution is limited. Although it proposes a bias-diversity decomposition of MBR, such decomposition is straightforward. The observations listed in Section 3.3 have mostly been covered in other literature, resulting in limited new insights.
>
> As explained in Line 145 to 147, the decomposition for the prediction of ensembled estimators shown by Krogh & Vedelsby, 1994 [1] differs from the well-known decomposition for the prediction of a single estimator shown by Geman et al., 1992 [2].
>
> - [1] Krogh, Anders, and Jesper Vedelsby. "Neural network ensembles, cross validation, and active learning." Advances in neural information processing systems 7 (1994).
> - [2] Geman, Stuart, Elie Bienenstock, and René Doursat. "Neural networks and the bias/variance dilemma." Neural computation 4.1 (1992): 1-58.
>
> Even though MBR decoding has a long history in NLP, nobody has found that it can be explained by the decomposition of Krogh & Vedelsby, 1994 [1]. We believe this viewpoint covers the empirically observed characteristics introduced in subsection 3.3. If you think this decomposition on the MBR decoding is straightforward and the novelty is limited, could you show papers explaining MBR decoding based on Geman et al., 1992 [1] or Krogh & Vedelsby, 1994 [2]?
>
> > The Performance of MAMBR with ancestral sampling/epsilon sampling/beam decoding is not significantly better than the baseline model, which weakens the paper's claim.
>
> As explained in subsection 3.3.2, our purpose is to show the exchangeability of the diversity of pseudo references and utility functions' output. Thus, if there are no changes between the results of MAMBR and that of diversified pseudo references, it supports our theoretical analysis. Therefore, if you feel there are no differences in the experimental results, it means that you recognize our experiment's successful results that show the exchangeability of the diversity of pseudo references and utility functions' output.
>
> ## Responses to Questions
>
> > For all three tasks, the utility function and the evaluation metric are the same. What will Figure 1 be if the utility function is different from the evaluation metric?
>
> We have explained the case using BLEURT, which is different from the utility functions, as the metric for measuring performance in Appendix D with Figure 3. This result shows similar tendencies to Figure 1.

---

> > ### Author Response · Authors · 2024-12-04
> > **Reminder**
> >
> > We believe that the reason you are not responding is not because you irresponsibly abandoned your role as a reviewer but because the concerns regarding this paper have been resolved. Therefore, if the concerns have indeed been resolved, we kindly request you to update your score regarding Soundness, Presentation, Contribution, and Overall Rating to reflect the outcome that no concerns remain, as is typically expected in the role of a reviewer after this discussion period.

---

### Official Review · Reviewer_bzKq · 2024-11-05

**Soundness:** 2
**Presentation:** 3
**Contribution:** 3
**Rating:** 5
**Confidence:** 3

**Summary:**

### tldr;

This paper provides valuable theoretical insights through its new framework decomposing bias and diversity in MBR decoding's quality estimation errors. The theoretical contribution is significant and well-presented. However, the empirical section needs improvement. The experimental results are unclear and would benefit from better explanation and presentation. I would appreciate if the authors could address the specific concerns raised in the Weakness section.

### Summary

- This paper presents a bias-diversity decomposition of the quality estimation error in Minimum Bayes Risk decoding
- The quality estimation error measures the discrepancy between the estimated quality using MBR decoding and the human judgement.
- Based on this decomposition the authors provide a novel theoretical interpretation of MBR decoding. -
	- The bias term captures the closeness between utility function and human evaluations i.e. the closeness between the human estimated quality and the quality estimated with a utility function.
	- The diversity term captures the variation in the estimated quality across different utility functions. Surprisingly this has a negative sign, suggesting that more diverse generations lead to lower error.
- Background in MBR:
  In MBR decoding the goal is not to find the most probable generation as is the case in typical beam-search decoding but rather find a generation that minimizes the expected risk for a given loss function and true posterior distribution
	- In practice, instead of expected risk, expected utility is computed by taking 1-risk. The expected utility is computed by comparing a generation to all other generation samples. Intuitively, MBR decoding ends up doing consensus decoding, i.e. picking generations which on average are most similar to other generations.

- The authors propose a new metric, pseudo-bias to approximate the bias term using gold-references since human judgements are expensive to obtain.
- The authors also propose a new MBR decoding approach called Metric-augmented MBR which increases diversity by adjusting the behavior of utility functions without altering the pseudo-references.
- In Section 3.3 the authors provide an interpretation of this theoretical decomposition and its relation to prior studies.

**Strengths:**

### Strengths
- The interpretation provided in section 3.3 is particularly nice as it provides additional evidence from prior work. I also like the explicit calling out of "Unexplored Aspect" which is lated discussed in section 4.2 and 5.4.
- I admire the scholastic writing rigor of the paper. The authors have generally cited prior work well going back decades, a practice which is commonly missing in recent ML literature. However, some more details on Minimum Bayes Risk Decoding could further strengthen the paper and make it self-contained.
- The results of experiment in Section 5.4 for adjusting the diversity term by varying model parameters and pseudo-references respectively are convincing

**Weaknesses:**

### Weakness
- Section 5.2:
	- Based on the theoretical decomposition a reader might expect the overall MSE to have highest correlation with performance; followed by overall bias and overall diversity. It is not clear as to why the one best bias or one best MSE correlates more strongly with performance than overall MSE. This does not provide a strong evidence for the theoretical decomposition presented.
- I found Figure 2 to be confusing. For e.g. consider this sentence "The results show that while ancestral sampling exhibits the highest bias, except in the case of the SAMSum dataset, it sometimes outperforms other sampling methods owing to its greater diversity." This statement is clearly false based on the plots.  For many plots bias and diversity move in the same direction as is predicted by the theoretical relationship but for many plots they move in opposite directions. For e.g. consider the plot for MSCOCO dataset. The overall bias and the one best bias go down while the overall diversity increases.
- Going beyond the math, at an intuitive level, it is difficult to interpret the diversity term. From my understanding, MBR decoding tries to select generations which on average are similar to most other generations. However, the diversity terms suggests that having diverse generations actually leads to better estimation of quality error and thereby better performance. Though MBR is trying to do something exactly opposite to this. It is difficult to reconcile this tension and some more discussion in building the intuition around this is lacking in the paper.

**Questions:**

Please refer to the concerns raised above.

---

> ### Author Response · Authors · 2024-12-01
> **Responses to Weaknesses**
>
> Thank you for reviewing our paper and sharing your feedback. We have updated our manuscript based on your opinion.
>
> >Section 5.2: Based on the theoretical decomposition a reader might expect the overall MSE to have highest correlation with performance; followed by overall bias and overall diversity. It is not clear as to why the one best bias or one best MSE correlates more strongly with performance than overall MSE. This does not provide a strong evidence for the theoretical decomposition presented.
>
> As explained in section 2, subsection 3.1, and lines from 151 to 153 in subsection 3.2, MBR decoding supports to estimate the precise quality of all candidates to choose the best one from them. Thus, both one best and overall MSE are important. However, since the final results of the MBR decoding are calculated by the one best result, One Best Bias is also important. We explained this finding in lines 420 to 422. To easily read the tendency of Figure 1, we have added the results of significance tests as underlined scores explained in the caption of Figure 1 and the paragraph named Settings in subsection 5.2. Moreover, instead of the previous setting All, we report more mathematically appropriate averaged correlations based on Fisher transformation [1] over all datasets as Avg. in Figure 1.
>
> [1] David M Corey, William P Dunlap, and Michael J Burke. Averaging correlations: Expected values and bias in combined pearson rs and fisher’s z transformations. The Journal of general psychology, 125(3):245–261, 1998.
>
> > I found Figure 2 to be confusing. For e.g. consider this sentence "The results show that while ancestral sampling exhibits the highest bias, except in the case of the SAMSum dataset, it sometimes outperforms other sampling methods owing to its greater diversity." This statement is clearly false based on the plots. For many plots bias and diversity move in the same direction as is predicted by the theoretical relationship but for many plots they move in opposite directions. For e.g. consider the plot for MSCOCO dataset. The overall bias and the one best bias go down while the overall diversity increases.
>
> First of all, as explained in Line 368 of subsection 5.2, lower bias and MSE are better for performance. To easily understand that, we have added an explanation about that in the caption of Figure 2.
>
> > Going beyond the math, at an intuitive level, it is difficult to interpret the diversity term. From my understanding, MBR decoding tries to select generations which on average are similar to most other generations. However, the diversity terms suggests that having diverse generations actually leads to better estimation of quality error and thereby better performance. Though MBR is trying to do something exactly opposite to this. It is difficult to reconcile this tension and some more discussion in building the intuition around this is lacking in the paper.
>
> To support the intuitive understanding, we have added explanations about the importance of diversity as lines from 205 to 208 in subsection 3.3.2 based on Jinnai et al. (2024) [2].
>
> [2]: Yuu Jinnai, Ukyo Honda, Tetsuro Morimura, and Peinan Zhang. Generating diverse and high-quality texts by minimum Bayes risk decoding. In Lun-Wei Ku, Andre Martins, and Vivek Sriku- mar (eds.), Findings of the Association for Computational Linguistics ACL 2024, pp. 8494–8525, Bangkok, Thailand and virtual meeting, August 2024a. Association for Computational Linguistics.

---

> > ### Author Response · Authors · 2024-12-04
> > **Reminder**
> >
> > We believe that the reason you are not responding is not because you irresponsibly abandoned your role as a reviewer but because the concerns regarding this paper have been resolved. Therefore, if the concerns have indeed been resolved, we kindly request you to update your score regarding Soundness, Presentation, Contribution, and Overall Rating to reflect the outcome that no concerns remain, as is typically expected in the role of a reviewer after this discussion period.

---

### Official Review · Reviewer_VnrG · 2024-11-07

**Soundness:** 2
**Presentation:** 3
**Contribution:** 2
**Rating:** 3
**Confidence:** 4

**Summary:**

This work aims at analyzing the characteristics of Minimum Bayes Risk (MBR) decoding through a bias-diversity/variance decomposition. On various tasks (e.g., MT, summarization, image captioning) and with a collection of sampling methods for generating pseudo-references, the authors show correlations between the bias, diversity, and overall MSE measures in MBR and the task performance. The authors argue for the importance of the diversity/variance measure in particular and propose a MAMBR approach that uses multiple loss functions to enhance MBR decoding.

**Strengths:**

This paper tackles an interesting problem of analyzing and enhancing MBR decoding. The experiments were conducted with multiple models, sampling methods, and tasks, ranging from pure text to multimodal scenarios. The writing is overall clear.

**Weaknesses:**

(1) Eq. (8) has counterintuitive namings. The L.H.S. of the equation corresponds more to the usual "bias" term and the current bias term on the R.H.S. corresponds more to the usual "MSE" term. Using the regular terminology, isn't the equation just MSE = bias^2 + variance?

(2) The work motivates with human-estimated quality (f hat) throughout the work, but uses a pseudo-bias that measures the metric/loss to gold references. The correlation between the pseudo-bias and actual human-estimated quality is not discussed.

(3) Meaningfulness of Figure 1. Since the pseudo-biases are direct functions of the gold references, the correlation between them and the task performance (also based on gold references) is less interesting. The diversity/variance measure can be interesting, but the correlation is also weaker. Also, for all of the reported correlation numbers, there are no associating p-values.

(4) Effectiveness of MAMBR. The proposed MAMBR simply uses multiple rather than one metric/loss in the MBR calculation. The results in Table 1, 2 and 3 consistently show no or extremely small improvement of MAMBR over regular MBR decoding.

**Questions:**

N/A

---

> ### Author Response · Authors · 2024-12-01
> **Responses to Weaknesses**
>
> Thank you for reviewing our paper and sharing your feedback. We have updated our manuscript based on your opinion.
>
> > (1) Eq. (8) has counterintuitive namings. The L.H.S. of the equation corresponds more to the usual "bias" term and the current bias term on the R.H.S. corresponds more to the usual "MSE" term. Using the regular terminology, isn't the equation just MSE = bias^2 + variance?
>
> As explained in Line 145 to 147, the decomposition for the prediction of ensembled estimators shown by Krogh & Vedelsby, 1994 [1] differs from the well-known decomposition for the prediction of a single estimator shown by Geman et al., 1992 [2].
>
> - [1] Krogh, Anders, and Jesper Vedelsby. "Neural network ensembles, cross validation, and active learning." Advances in neural information processing systems 7 (1994).
> - [2] Geman, Stuart, Elie Bienenstock, and René Doursat. "Neural networks and the bias/variance dilemma." Neural computation 4.1 (1992): 1-58.
>
> > (2) The work motivates with human-estimated quality (f hat) throughout the work, but uses a pseudo-bias that measures the metric/loss to gold references. The correlation between the pseudo-bias and actual human-estimated quality is not discussed.
>
> To deeply understand how the pseudo-bias correlates to human evaluation, we have added the actual correlation of the metrics we used to calculate the pseudo-bias into the footnote of subsection 4.2. As explained in this part, the Pearson correlations of COMET (Unbabel/wmt22-comet-da) and BERTScore with microsoft/deberta-xlarge-mnli are 0.990 on the system-level task for English to German [3] and 0.7781 (https://github.com/Tiiiger/bert_score) on WMT16 to English [4], respectively.
>
> - [3] Freitag, Markus, et al. "Results of WMT23 metrics shared task: Metrics might be guilty but references are not innocent." Proceedings of the Eighth Conference on Machine Translation. 2023.
> - [4] Bojar, Ondřej, et al. "Results of the wmt16 metrics shared task." Proceedings of the First Conference on Machine Translation: Volume 2, Shared Task Papers. 2016.
>
> > (3) Meaningfulness of Figure 1. Since the pseudo-biases are direct functions of the gold references, the correlation between them and the task performance (also based on gold references) is less interesting. The diversity/variance measure can be interesting, but the correlation is also weaker. Also, for all of the reported correlation numbers, there are no associating p-values.
>
> To clarify how the correlations are significant, we have conducted the statistical significance test and underlined the correlation scores with statistical significance with p<0.05. Furthermore, instead of the previous setting All, we report more mathematically appropriate averaged correlations over all datasets as Avg. in Figure 1 using Fisher transformation [5]. Corresponding to these modifications, we have added further explanations to the caption of Figure 1 and the paragraph named Settings in subsection 5.2.
>
> [5] David M Corey, William P Dunlap, and Michael J Burke. Averaging correlations: Expected values and bias in combined pearson rs and fisher’s z transformations. The Journal of general psychology, 125(3):245–261, 1998.
>
> > (4) Effectiveness of MAMBR. The proposed MAMBR simply uses multiple rather than one metric/loss in the MBR calculation. The results in Table 1, 2 and 3 consistently show no or extremely small improvement of MAMBR over regular MBR decoding.
>
> As explained in subsection 3.3.2, our purpose is to show the exchangeability of the diversity of pseudo references and utility functions' output. Therefore, if there are no changes between the results of MAMBR and those of diversified pseudo references, this supports our theoretical analysis and shows the success of our experimental results.

---

> > ### Author Response · Authors · 2024-12-04
> > **Reminder**
> >
> > We believe that the reason you are not responding is not because you irresponsibly abandoned your role as a reviewer but because the concerns regarding this paper have been resolved. Therefore, if the concerns have indeed been resolved, we kindly request you to update your score regarding Soundness, Presentation, Contribution, and Overall Rating to reflect the outcome that no concerns remain, as is typically expected in the role of a reviewer after this discussion period.

---

### Meta-Review · Area_Chair_Uzfx · 2024-12-19

**Metareview:**

The authors study the MBR decoding objective, comparing the objective from maximizing the pseudo-reference similarity to a human-estimated ground truth. The core object of study is a bias-diversity term (not to be mistaken with bias variance..) where the bias term corresponds to the per-sample error between the similarity f and the human estimate \hat{u} and the diversity term is the same thing but with \bar{u} the mean.

The paper studies an important problem (MBR decoding, and understanding when and why such techniques help), and the experimental coverage is nice and extensive.

As reviewers note, however, the theory part doesn't seem quite as strong (which especially seems important given the title). Reviewer VnrG is not right that this is exactly bias-variance, but they are right that the term that's named "bias" is really much closer to a traditional MSE term. The 'diversity' term here is really showing that you need to match the variance you see in the MSE term (otherwise you're not distributionally matching u, instead you might be matching only the mode) and this leads to some counterintuitive confusion like the one mentioned by reviewer bzKq. Finally, I don't necessarily think reviewer zsMt is right that this exact result appears in other works, but I think the argument that the decomposition itself is technically simple (appendix A and B are fairly standard manipulations of a quadratic sum) is right, and that the paper can't quite argue that the decomposition itself is a major technical achievement.

**Additional Comments On Reviewer Discussion:**

The authors made some writing changes during the rebuttal, though I think the main complaints placed by the reviewers that I saw were closer to more of the underlying framing of the bias-diversity tradeoffs posed by the authors.

---

### Decision · Program_Chairs · 2025-01-22

Reject